# Characteristics of Graphene Oxide for Gene Transfection and Controlled Release in Breast Cancer Cells

**DOI:** 10.3390/ijms23126802

**Published:** 2022-06-18

**Authors:** Francesca Grilli, Parisa Hajimohammadi Gohari, Shan Zou

**Affiliations:** 1Metrology Research Centre, National Research Council of Canada, 100 Sussex Drive, Ottawa, ON K1A 0R6, Canada; francesca.grilli@nrc-cnrc.gc.ca (F.G.); parisa.gohari@nrc.cnrc.gc.ca (P.H.G.); 2Ottawa-Carleton Institute for Biomedical Engineering, University of Ottawa, 800 King Edward Avenue, Ottawa, ON K1N 6N5, Canada; 3Department of Chemistry, Carleton University, 1125 Colonel By Drive, Ottawa, ON K1S 5B6, Canada

**Keywords:** controlled release, cytotoxicity, surface functionalization, pH/redox-dependent, drug delivery

## Abstract

Functionalized graphene oxide (GO) nanoparticles are being increasingly employed for designing modern drug delivery systems because of their high degree of functionalization, high surface area with exceptional loading capacity, and tunable dimensions. With intelligent controlled release and gene silencing capability, GO is an effective nanocarrier that permits the targeted delivery of small drug molecules, antibodies, nucleic acids, and peptides to the liquid or solid tumor sites. However, the toxicity and biocompatibility of GO-based formulations should be evaluated, as these nanomaterials may introduce aggregations or may accumulate in normal tissues while targeting tumors or malignant cells. These side effects may potentially be impacted by the dosage, exposure time, flake size, shape, functional groups, and surface charges. In this review, the strategies to deliver the nucleic acid via the functionalization of GO flakes are summarized to describe the specific targeting of liquid and solid breast tumors. In addition, we describe the current approaches aimed at optimizing the controlled release towards a reduction in GO accumulation in non-specific tissues in terms of the cytotoxicity while maximizing the drug efficacy. Finally, the challenges and future research perspectives are briefly discussed.

## 1. Introduction

Breast cancer is one of the most frequent malignant tumors that affect women globally. Even though standard breast cancer treatments, including surgery, radiotherapy, and chemotherapy, have been relatively effective, significant limitations and side effects still jeopardize the quality of patients’ lives [1]. There are currently two FDA-approved drugs to alleviate the side effects of chemotherapy, Doxil and Abraxane, which are based on doxorubicin (DOX) and paclitaxel, respectively [2]. However, the detrimental effects of chemotherapy strategies on normal tissues due to the multidrug resistance (MDR) effect and high concentration of the drug used cannot be eliminated [3]. In order to address the shortcomings of chemotherapy, breast cancer treatment strategies based on polymers [4,5,6], dendrimers [7,8], carbon-derivatives [9,10], and liquid metal and metal–organic framework nanoparticles [11,12] have been developed to deliver small drug molecules, antibodies, and nucleic acids to the targeted sites [13].

Over the years, gene transfection by carriers has become an alternative treatment because it is based on inducing or replacing overexpressed genes or proteins in tumor cells, leading to cancer cell death and a reduction in tumor growth [13]. Among these carriers, GO has recently drawn the interest of researchers due to its unique properties, including its controllable 2-dimensional structure, high surface area, and versatile functional groups. Interestingly, these features not only give this material the capability of being rationally functionalized, but also to respond to the stimuli from its surrounding environment. Thus, these characteristics provide GO with the distinctive capability to be tailored in multifunctional nanocarriers for various applications [14]. Nonetheless, engineering a drug/gene carrier that can ideally meet all of the essential requirements of a delivery system and finally reach clinical trials is still challenging. According to the existing definitions [15], a potential drug/gene carrier must be capable of (1) protecting the drug/gene from degradation upon transfer; (2) retaining it in the circulation system for more than a few hours; (3) facilitating cell internalization and drug release at the desired sites; (4) being non-toxic and biodegradable. All of the mentioned criteria are highly dependent on the carrier’s physicochemical properties (such as its morphology, including shape, size, and aspect ratio, ionic strength, and charges, and its chemical functionality), determining its potential to pass through the biological compartments. Furthermore, the chemical modification of GO provides the opportunity to improve the performance of GO-based nanocarriers in terms of their toxicity, specific targeting, cell internalization, and drug release. The conjugation of these biocompatible materials enables the carrier to associate with nucleic acids, and at the same time preserves them from cleavage in biological fluids [16]. 

Once the carriers are internalized into cells, gene derivatives are released in the desirable cell compartment. Subsequently, the nucleic acids can direct the formation of the mutated genes, with the alternation of their nature resulting in the expression or inhibition of specific proteins. Obviously, this ability to manipulate gene expression or inhibition is highly affected by the release of gene derivatives from the carriers [17]. Taking advantage of the readiness for GO functionalization, this material is responsive to stimuli such as pH, temperature, and near-infrared irradiation (NIR) [18]. This susceptibility and the possibility of being functionalized with stimuli-responsive agents through specific linkages can lead to multi-responsive GO carriers being engineered for a controlled and efficient release.

In this review, we describe the characteristics of GO and functionalized GO as nanocarriers, their modification for nucleic acid delivery, their toxicity, and their controlled release of payloads with the aim of providing a breast cancer treatment by summarizing the most recent publications (before October 2021). The following combination of words, such as GO, modification, toxicity, gene/drug deliver, controlled release, and breast cancer, were applied when searching papers using the Science Direct and PubMed engines.

## 2. Characteristics of GO as a Nanocarrier

Among all nanocarriers, carbon-based materials display various structures, such as a tetrahedral *sp*^3^ network, planar *sp*^2^, and linear *sp* configurations [19], which result in the presence of allotropes, including 0-dimensional (0D) fullerene, 1-dimensional (1D) carbon nanotubes, and 3-dimensional (3D) graphite [20]. As a carbon derivative, graphene comprises a monolayer of *sp*^2^ carbon atoms aligned in a honeycomb-like two-dimensional (2D) network. Despite its optical properties and large surface area, the hydrophobic nature of graphene has caused aggregations that have hindered its functionality in solution-processed applications. In order to overcome this limitation, the synthesis of GO and reduced GO (rGO) was introduced and has improved the solubility and processability of graphene-based materials [21,22]. In comparison with other carbon-based materials, GO has a large surface area with theoretical and measured values of around 2418 and 2391 m^2^/g, respectively [23], an order of magnitude more than that of the majority of other nanomaterials [24]. Unlike the hydrophobic nature of graphene, GO has both hydrophobic regions and hydrophilic peripheries, giving this material an amphiphilic character [25]. This composition is responsible for the good aqueous dispersibility and the exceptional ability to cross cell membranes easily. These unique structures allow biological molecules to be associated with GO flakes, facilitating the efficient loading of drugs and genes or a combination of them. 

The synthesis of GO has developed over the years [26,27,28] after the original protocol was proposed [29]. Recently, the modified Hummers method has been used to produce exfoliated graphene oxide flakes with the potential for additional modification in biomedical applications [22,25,28,30,31]. As a result, graphene is highly oxidized, leaving hydroxyl and epoxy functional groups on the planar surface, while the carboxylic groups are found at the edges [32]. Depending on the post-processing step after the synthesis and purification, a wide range of exfoliation [25] and oxidation degrees can be obtained, with the possibility for size distribution control [33]. 

### 2.1. Size and Two-Dimensional Structure

When nanoparticles are in the body, they must pass through several barriers to reach the targeted site and successfully release the therapeutic chemicals into the appropriate biological compartment. The journey of the nanoparticle initiates within the blood vessels, and if the targeted cells are out of blood circulation (e.g., solid tumors), the nanoparticle encounters three primary barriers prior to reaching the malignant tissue: (1) the endothelium of the vessel walls; (2) the extracellular matrix (ECM); (3) the cell membrane [34]. In the case of breast cancer therapy, the carrier needs to pass through the mentioned barriers, but it should not penetrate the blood–brain barrier (BBB) to avoid potential brain toxicity [35]. The BBB separates the brain from the blood circulation with a continuous layer over the thick basement membrane.

The size of the drug carriers determines their potential to pass through the biological compartments. Larger carriers (in the micron range) are likely to be cleared by reticuloendothelial macrophages through phagocytosis before reaching the targeted site of the body. Carriers at the nanoscale can circulate in the body for a more extended period without being recognized by macrophages as invaders [36,37,38,39]. Specifically, small carriers (<200 nm) are required for endothelial penetration when the targeted tissue is positioned outside the bloodstream [15]. The size of the endothelial fenestrations for carrier penetration relies on the pathological conditions that the carrier is dealing with [40]. Almost all solid tumors have leaky angiogenesis and impaired lymphatic function, differentiating them from other tissues. A growing tumor creates an excessive network of blood vessels to increase oxygen and nutrition transfer. This property allows particles of 10 to 100 nm in size to pass through the voids of blood vessel walls and to accumulate in tumor locations instead of normal tissues, which is known as enhanced permeability and retention (EPR) [41], while it was also reported that even larger carriers (200−800 nm) could penetrate through the endothelium barrier and accumulate in the targeted tumor [42]. These results indicate the easier and more efficient diffusion of smaller particles (<100 nm), while the penetration of the larger ones (<800 nm) is not entirely prevented. Subsequent to successfully passing through the endothelium, the nanoparticle should be capable of going through the collagen fibers, which is the primary component of the ECM. The gaps between the fibrils and the intra-fibrillar spacing are 20−40 and 75−130 nm, respectively [43]. For this reason, the ideal size of a carrier was reported to be less than 200 nm to not only prevent its clearance from the circulation [44], but also to facilitate its diffusion throughout the ECM. All of the reported information regarding nanoscale carriers raises the demand for an appropriate range of carrier sizes as one of the criteria to achieve adequate drug or gene transportation. Therefore, a material with unique size and dimension characteristics is more advantageous as a drug or gene carrier.

Compared to all of its counterparts, GO can meet the required criteria mentioned above due to its tunable lateral size from a few nanometers to tens of microns [45,46] and its monolayer thickness at the sub-nanometer scale. The reported thickness (vertical size) of a GO monolayer ranged from approximately 0.5 to 1.2 nm (measured by atomic force microscopy topography imaging), while larger reported scales are associated with multilayer GO [28,47,48]. On the other hand, the wide range of lateral sizes is advantageous to meet different requirements for biological applications [49,50]. GO nanoparticles ranging from 50 to 200 nm have consistently been demonstrated in multiple investigations to bypass immune cells, diffuse through vessel walls and the ECM, and infiltrate breast cancer cells effectively [51,52,53,54,55].

GO cell penetration occurs through three different mechanisms depending on the carrier size. Smaller particles, ranging from 20 to 100 nm, internalize cells through caveolin-dependent endocytosis, while clathrin-dependent endocytosis occurs for larger particles of 100–500 nm. Additionally, carriers possessing a much larger size (0.5−1.5 µm) are absorbed by micropinocytosis [34]. According to the research findings, GO nanoparticles are more likely to be taken up by cancer cells via clathrin-mediated endocytosis than other nanoparticles because of the planar 2D shape of the GO flakes, which possess sharp edges [56,57]. 

According to the literature, various approaches have been conducted to obtain GO with specific lateral dimensions applicable before, during, or after the synthesis. The approaches used prior to the synthesis include adopting a suitable graphite as the starting material with a suitable particles size, or increasing the mass of the oxidative agent and the time of oxidation, while post-processing methods are based on ultrasonication, centrifugation, and dialysis/filtration [33]. These strategies cause GO breakage, which initiates from defected sites with sp^3^ bonds. Consequently, if the size reduction procedure continues, it will ultimately lead to the removal of functional groups at the edges. By applying a range of ultrasonication energy, the diameters of two types of commercial graphene oxide flakes could be tuned from about 2 microns to around 150 nm [58]. In another study, utilizing the centrifugation of 0.01 mg/mL GO in a mixture of water and glycerol at 8000 g for 3 h could decrease the GO flake size to 150−850 nm [59], as shown in Figure 1.

It was also reported that the lateral width of GO is highly influenced by the initial size of the raw graphite flakes used for synthesis. By breaking down graphite as a starting material, GO with a 10−300 nm lateral size can be obtained [60]. Using graphite nanofibers with 130 nm diameter as a precursor, Luo et al. could scale down the average lateral width of GO flakes to 100 nm for cancer drug release [61]. These results indicated that depending on the application, fabricating a carrier that can diffuse the mentioned barriers to reach the targeted site of the body raises the demand for achieving a uniform size of GO particles using the mentioned approaches. In addition, different sizes of functionalized GO have been utilized for breast cancer therapy [17,51,53,62,63,64,65,66,67,68,69].

Ultrasonication has been used not only to reduce the flake size to 30−500 nm, but also to exfoliate graphite into monolayer GO. Furthermore, increases were found in the average lateral dimension and height of GO after functionalization, which were considered the result of successful modifications [51,62,63,64,65,66,67,70,71,72]. Although the mentioned methods reduce the lateral dimension of GO, it should be noted that the ultimate nanoparticle that interacts with breast cancer cells has a different size compared to GO alone, depending on the amount of functionalization and the payload mass [51,62,63,64,65,66,67,70,71,72].

As explained above, an ideal carrier needs to have a long blood circulation time and bypass the phagocytosis by macrophages to reach the targeted site, and at the same not diffuse past the blood–brain barrier [35]. Zhang et al. focused on the effect of lateral dimension of GO on the blood circulation time and clearance by macrophages. They synthesized GO particles ranging from 10−800 nm and evaluated the blood circulation times in vivo. It was shown that considering the wide size distribution, GO particles with a larger size are likely to be eliminated more readily, and the half-life of the blood circulation was reported to be about 5.3 h, which was higher than for single-walled carbon nanotubes and fullerene, according to previous findings [73,74,75,76]. This suggested that GO can be a promising candidate for gene delivery due to its distinctive size characteristics. Additionally, the polymer modification of GO was shown to slow down phagocytosis. Other parameters, such as the cell type, dosage, and exposure time, are also involved in the clearance and cellular uptake of GO, which will be discussed in the next section. 

### 2.2. Functionalization

GO can be considered as a form of functionalized graphene, encompassing abundant oxygen-containing groups, which enable the nanocarrier to be specifically modified and loaded with therapeutic agents [77,78]. With the polar basal plane and hydrophilic -OH and -COOH groups, GO is dispersible in water, similar to an amphiphilic molecule that can be used as a surfactant to stabilize hydrophobic species in water (e.g., drugs) [79,80].

GO typically has a negative surface charge when dispersed in water, mainly due to the ionization of the carboxylic acid and hydroxyl groups. This negative charge could provide electrostatic repulsion, allowing a stable GO dispersion. The ability of GO to disperse in aqueous environments has been demonstrated as an advantage for targeting and release mechanisms and imaging in cancer therapy [63,81,82].

GO is more hydrophilic in acidic environments, affecting GO suspensions’ zeta potentials [63,79,80]. Alkaline pH causes the ionizable groups (carboxylic and hydroxyl groups) of GO to dissociate, resulting in a greater negative charges [81,82]. This suggests that GO can be made into a smart drug delivery system due to its controlled release properties in diverse biological environments by fine-tuning its one-of-a-kind pH characteristics. Those functional groups are in fact highly affected by the pH level of the surrounding medium due to the affinity to accept or give out protons. The hydrophilicity of GO increases with pH due to the protonation of carboxyl groups on GO [81,82].

Even though GO disperses in cell culture and physiological environments due to its hydrophilic characteristics, in order to avoid aggregation and overcome the π-π stacking and van der Waals forces between sheets, it is necessary to functionalize it through various methods to sustain its single-layer structure [83]. These modifications are categorized into covalent and non-covalent methods [84,85], which are utterly dependent on the loading strategy of the drug or gene, as summarized in Table 1.

#### 2.2.1. Covalent Modification

Polymer materials, such as polyethylene glycol (PEG) [86], polyethyleneimine (PEI) [92], polyvinyl alcohol (PVA) [93], dextran (DEX) [90], polyacrylic acid (PAA) [94], and chitosan [95], have been covalently bonded to enhance the biocompatibility and stability of GO-based delivery systems. Covalent modification routes are generally carried out using three different reactions: (1) a ring-opening reaction of epoxide groups present on the basal plane of GO via nucleophilic attacks; (2) amidation, where COOH groups at the edges of GO flakes are replaced by NH_2_ in polymer groups; (3) esterification, in which carboxylic groups of GO directly react with hydroxyl groups of the polymer in the absence of NH_2_ as a mediator agent [17,77,91], as illustrated in Figure 2.

The most commonly utilized polymers to assist in the loading of genes for breast cancer treatment include PEG [17,62,88], polyamidoamine (PAMAM) [16,17,87], PEI [88], and chitosan [64,65,66]. Due to the negative charges of GO and gene derivatives, these cationic mediators form a stable interaction between the GO and the genes. To covalently modify GO with these polymers, 1-ethyl-3-(3-dimethylaminopropyl) carbodiimide (EDC) and N-hydroxysuccinimide (NHS) were used as key agents to expose reactive C=O in carboxylic groups of GO to form amide bonds with primary amine (NH_2_) moieties. The reason for using both agents is that when EDC reacts with a carboxyl group of GO, it will generate amine-reactive O-acyl isourea. If this product does not interact with an amine, it will hydrolyze and renew the carboxylic acid. NHS is added to generate amine-reactive NHS esters, leading to a more stable amide bond between the polymer and GO [96] (Figure 2). In some studies, all epoxide groups of GO have been altered to carboxylic groups to increase the polymer modification, thereby generating a more positive charge through the primary amine content for higher gene loading efficiency [64,65]. It is worth noting that the amide bonds formed between −COOH and the amine-terminated agents are sensitive to pH alterations. In acidic environments such as cancerous cells, the amide bond weakens, leading to the dissociation of the cationic agent containing the payload from GO because of protonation [17,64,97,98]. This shows the essential role of GO functionalities in delivering genes to the desired cell compartment.

Among these polymers, PEG has attracted particular interest for biocompatibility enhancement because it increases the blood circulation time without being recognized by macrophages and alleviates GO particle aggregation in the biological medium [21]. Subsequent to PEGylation, GO is functionalized with other polymers, such as PEI and PAMAM, to load gene derivatives for breast cancer treatments to enhance the gene loading efficiency due to their outstanding capability for encapsulating gene derivatives [17,67,88]. It was shown that PEI had an essential role in binding siRNA to cell-penetrating peptide (CPP)-modified GO-PEI-PEG. By increasing only the GO/PEI mass ratio from 60 to 100 in the carrier system, the complete siRNA protection by the carrier and its binding were confirmed by gel retardation assay [67]. This might be due to PEI’s ability to alter the surface charge of GO to positive values to a certain extent, since it has been reported that PEI could increase the zeta potential of GO to +47.2 mV due to its primary amines, while this value for bare GO was −28.9 mV [99]. The results demonstrated the same trend for PAMAM in terms of the zeta potential, except that PAMAM is a dendrimer and its amine functionalities depend on the number of its generations, defined as the number of reaction sequences. As a result, its branched structure plays an essential role in encapsulating drugs and genes [17,100]. Yadav et al. [17] reported that PEGylation was necessary for the carrier’s stability, as bonding PAMAM individually to GO causes precipitation. The presence of two or more polymers along with PEG improves the drug or gene encapsulation and provides protection against enzyme-induced degradation. Chen et al. [64] demonstrated that the presence of chitosan in the GO-based carrier could efficiently protect siRNA from degradation. Likewise, Wang et al. [51] showed that bare GO could not retard the migration of siRNA, and the GO modification with octaarginine (R8), a cell-penetrating peptide, was essential to protect the payload.

It should be noted that in some studies, PEGylation was carried out through the ring-opening reaction of epoxide groups and nucleophilic attacks on carbonyls [17], while in other studies the amidation strategy was used to modify GO with polymers and peptides [62,67,88].

#### 2.2.2. Non-Covalent Modification

In addition to covalent modifications, hydrogen bonding, π-π stacking, and electrostatic interactions were the major driving forces for small drug molecules [16,65], phospholipids [70], and oligonucleotides [68,72] to interact with GO, suggesting that gene derivatives can be directly loaded on GO.

The aromatic rings that exist in the bases of genes enhance such short-range π-π interactions, which are sensitive to the pH level of surrounding medium. It was reported that miRNAs, siRNAs, and some short-stranded DNA strands dissociate from GO when the medium becomes acidic. This might weaken the π-π stacking interactions [101], which is beneficial to the cancerous cells, where the environment is often acidic. These forces are not only limited to gene derivatives but to most drugs, including DOX, Dinaciclib, and Xanthohumol, which have also been loaded on GO through the same interactions for breast cancer treatment [65,86,88].

In addition to pH, these aromatic C-C bonds can respond to NIR irradiation. Pulsed laser irradiation NIR activates photosensitizing agents with an affinity for optical wavelengths. When the NIR light or photons interact with GO, this can lead to a forced resonance vibration mechanism, resulting in heat production at around 50 °C. This phenomenon has been demonstrated to weaken π-π interactions between drugs and genes by applying an NIR laser at 808 nm [62,66,102,103,104]. The effect of pH and NIR irradiation on the release is described thoroughly in Section 4.

On the other hand, the challenges associated with transferring naked nucleic acids to the targeted cells are their negative charge, high molecular weight, enzymatic degradation, instability, and low cellular uptake [51,62]. Therefore, most nucleic acids such as plasmid DNA, DNA aptamers, siRNAs, and miRNAs transfected to cancer cells are loaded on GO via intermediaries through electrostatic interactions. As a result, the surface charge (zeta potential) is altered to cationic (positive), promoting electrostatic interactions with anionic oligonucleotides [105].

Although polymers and all materials with a positive charge interact with GO through electrostatic interactions [67,70], covalent modifications are more desirable due to their stronger bonding, protecting the GO carrier system from degradation before reaching the targeted site of the body [16,17,62,63,77,86,87,88,89].

#### 2.2.3. Targeting Strategy

After bypassing macrophages and degradation enzymes of the body, the carriers must reach the desired region of the body. The recruited strategies for a carrier to find its way to the cancerous part are active and passive targeting. The former uses EPR to reach malignant tissues, while the latter is based on the ligand–receptor approach to internalize individual cancer cells [106]. Although GO carriers can be internalized in cells depending on their size, shape, charge, and functional groups, the presence of an active targeting agent can further assist the carriers to distinguish breast cancer cells from healthy ones. 

Similar to other cancers, breast cancer cells produce overexpressed proteins or receptors, such as glycosaminoglycans (GAGs) [62], hyaluronic acid (HA) [65], and folic acid (FA) [66] binding receptors on the cell membrane. In addition, there are also HER2 proteins, which are responsible for the growth and division of healthy breast cells but are overexpressed on the membranes of cancerous ones [107]. Epithelial cell adhesion molecule (EpCAM) is also another transmembrane glycoprotein, which is overexpressed in cancerous epithelial cells with fast proliferation and can be found in breast cancer cells [64]. These proteins can be directly recognized by targeting molecules added to GO-based carriers. After being combined with polymers or biocompatible agents, the covalent or non-covalent modification of GO with peptides, antibodies, aptamers, or hyaluronic and folic acids enhances its cell penetration potential to target the specific breast cancer cells. Table 2 represents the GO modifications, payload types, targeting agents, and breast cancer cell types. 

Covalent modifications are the most common strategies to bind targeting agents to functional groups on GO through the formation of amide bonds between carboxylic groups and the amine moieties. Nonetheless, these modifications were not limited to covalent bonding and GO could form different kinds of interactions with targeting agents [72].

In recent studies, GO was covalently linked to cell-penetrating peptides such as poly-L-arginine (PLA) and octaarginine (R8) through an amidation reaction [51]. On the other hand, another study applied electrostatic interactions for R8 modifications [67]. Whether using covalent or non-covalent interactions, these studies have shown that the negative zeta potential values of the GO-based carries were altered to positive values, resulting in tunable gene–carrier interactions. The presence of arginine in peptides plays an essential role in cell internalization, and the guanidine groups are vital for the targeting of cancer cells, which interact with overexpressed GAGs on cancer cell membranes [51,62,67,70].

Similarly, aptamers, such as MUC1 and AS1411, have also been used to functionalize GO to target breast cancer cells [72,90]. MUC1 has a high affinity to Mucin MUC1 glycoproteins that are overexpressed on MCF-7 and MDA-MB-231 cells. The MUC1 aptamer has been immobilized on GO along with the cytochrome C aptamer through π-π stacking interactions. In addition to the synergic effect of MUC1 on targeting, none of the aptamers has been demonstrated to migrate from GO due to the strong π-π stacking interactions with GO. This indicates the ability of GO to protect aptamers in the absence of any other mediator [72]. Unlike MUC1, the synthetic 26-base DNA aptamer AS1411 [108] was conjugated to GO through hydroxyl groups of DEX and showed affinity towards nucleolin proteins on the 4T1 and MCF-7 cells [90].

On the other hand, antibodies such as anti-HER2 and anti-EpCAM have been associated with GO and interacted explicitly with overexpressed HER2 and EpCAM, respectively. The former was bound to GO through amidation in hydroxyl groups, while the latter was immobilized on GO through electrostatic interactions. These receptors were considered to be responsible for the fast proliferation of MCF-7 cells [51,64]. In addition, Antwi-Boasiako et al. [109] developed a GO-based Raman probe, which was tested on SKBR3 breast cancer cells with the overexpression of HER2. The results demonstrated that bioconjugated graphene oxide can selectively recognize cancer cells. Another targeting agent HA was covalently bonded on GO and showed an improved targeting efficiency to 4T1 breast cancer cells through its interaction with CD44 receptors [65,71]. At the same time, the presence of FA enhanced the targeting effect on MCF-7 and MDA-MB-231 cells through an interaction with folate receptors [63,66,88].

These targeting strategies of GO have also been adopted in biosensing to detect breast cancer cells, in which protein receptors can bind to GO through electrostatic and hydrophobic interactions. As an example, for the early detection of breast cancer, ERBB2, human epidermal growth factor receptor 2 was functionalized through EDC-NHS chemistry on the surface of a graphene foam modified with electrospun carbon-doped titanium dioxide nanofibers (nTiO2) and GO–gold nanoparticles. The results showed that these sensors have the potential to detect specific biomarkers in breast cancer cells [110,111].

#### 2.2.4. Gene Loading Efficiency of GO-Based Carriers

The loading efficiency of GO carriers depends on the ability of the whole gene carrier system to protect and simultaneously deliver genes. This ability is directly influenced by the type, structure, concentration, and magnitude of positive charges of modified GOs, which affects the gene silencing efficiency in breast cancer cells. In order to engineer an efficient GO-based gene carrier, optimal concentrations of the modification agents should be selected to completely preserve genes. The protection level of GO carriers to genes has been determined by using gel retardation assays. In these tests, different nanocarrier/gene weight ratios were used to investigate the optimal ratio at which the carrier can retard gene migration [51,62,64,70]. Table 2 shows a summary of the most recent studies based on GO modification approaches, as well as the N/P ratio, in which N defines the nitrogen content of the amine-terminated polymers and peptides, while P presents the phosphorous present in the nucleic acids. 

PEGylation is a well-known GO modification approach due to its ability to make GO flakes stable in biological dispersions. It assists GO carriers in bypassing the body’s immune system [70]. The colloidal stability of GO-PEG in aqueous solution is due to the steric hindrance of PEG’s hydrophilic chains and the repulsion between its amine groups [70]. For this reason, amine–PEG has been used to functionalize GO with the purpose of delivering various miRNAs and siRNAs to breast cancer cells, including: (1) miRNA 101 and Rictor siRNA, which are regulators of the PI3K/AKT pathway and responsible for cell metabolism and proliferation; (2) EPAC1 siRNA to inhibit EPAC1 protein, which is responsible for cell metastasis; (3) cell death (cd) siRNA to induce apoptosis; (4) P-gp siRNA to inhibit multidrug resistance protein (P-gp) [17,62,67,70,88]. Despite its capability, PEG is commonly combined with other polymers to functionalize GO. It has been shown that PEG maintained the stability of gold nanoparticles associated with GO for four weeks without any aggregation. The results showed that the zeta potential of the gold nanoparticle–GO shifted to negative with PEGylation and could only retard miRNA 101 at the GO–carrier/miRNA ratio of 3.8. On the other hand, adding PLA to the carrier could alter the negative charge of the carrier to positive and retard the migration of miRNA at all GO–carrier/miRNA ratios lower than 3.8 (e.g., 2.8, 1.5, and 1). As a result, the release of miRNA-101 in MCF-7 and MDA-MB-231 cells could induce apoptosis by activating apoptotic pathways after 24 h, and it decreased the cell viability of MCF-7 and MDA-MB-231 to 62% and 64% with gold nanorods and nanoparticles, respectively [62].

In addition, Yadav et al. [17] reported that in their GO-based carrier functionalized with the dendrimer PAMAM, PEGylation was necessary to avoid precipitation. As mentioned earlier in Section 2.2.1, the branched structure of PAMAM can bring about a large number of positive charges on GO, followed by gene encapsulation. As a result, the efficient loading of EPAC1 siRNA on GO-PEG in the presence of PAMAM could lead to 62% gene silencing in MDA-MB-231 cells in vitro. These results indicated that even if PEGylated GO maintained the stability of GO, the presence of another mediator on GO was essential to form a sufficient electrostatic interaction, and thereby effective gene loading. A similar approach was used by Zeng et al. [88] to load P-gp siRNA on GO functionalized with both PEG and PEI, where PEI assisted GO-PEG in protecting the siRNA with its rich amine groups. The results showed that combining the siRNA and DOX with GO-PEG-PEI can increase protein inhibition by 70%.

On the other hand, there are studies that have focused on loading gene derivatives and functionalizing GO with PAMAM in the absence of PEG. PAMAM has been grafted on GO to deliver MMP-9 shRNA and miR-21i to breast cancer cells [16,87]. MMP-9 shRNA is a short hairpin RNA responsible for cancer metastasis, while miR-21i is an inhibitor of miR-21 with the capability for preventing cancer cell growth [16,87]. 

To prevent the overexpression of Survivin protein, an inhibitor of apoptosis, Survivin siRNA, was delivered to breast cancer cells through GO-modified chitosan and R8 [51,64]. Both studies showed that according to the Tyndall phenomenon, the covalent modification of GO with either chitosan or R8 increased the carrier stability in aqueous solutions, which might have been due to the electrostatic shielding effect. Their result demonstrated that GO–chitosan and GO–R8 loaded with Survivin siRNA induced 44% and 50% inhibition of the related-protein expression in vitro and in vivo, respectively. Therefore, the polymer modification was essential, and a 30:40 (*w*/*w*) N/P ratio was required to protect the siRNA in the GO carrier system [51,64]. 

Figure 3A illustrates the capability of the GO/R8/anti-HER2/Survivin-siRNA nanocarrier to inhibit the relative protein expression of MCF-7 cells. The same phenomenon can be observed for MMP-9 shRNA loaded on GO modified only with PAMAM, where 10 mg of the carrier system was required to deliver 1 mg of the shRNA in breast cancer cells in vitro and in vivo to cause 52% gene silencing (Figure 3B) [87]. Additionally, Izadi et al. [65] and Maheshwari et al. [66] reported that the functionalization of GO with trimethyl chitosan (TMC) and chitosan oligosaccharide changed the zeta potentials of GO from −29 to +38 mV and from −43 to +31 mV, respectively. This result was beneficial for the homogenous dispersion and effective interactions with gene derivatives. The combination of GO-TMC loaded with Dinaciclib and HIF-1α siRNA, which is defined as a hypoxia-inducible factor, resulted in 62% gene silencing in 4T1 cells in vitro. Moreover, this approach could protect the siRNA for 18 h, while GO grafted with chitosan oligosaccharides loaded with EGFR siRNA caused protein inhibition in MCF-7 and MDA-MB-231 cells. Other studies have reported on the delivery of siRNAs using R8-grafted GO, and the links were through either amidation or electrostatic interactions, which provided sufficient positive charges with its -NH2 terminals for siRNAs to be absorbed [51,67,70].

Among gene derivatives that have been loaded on GO carriers for breast cancer treatment, fewer studies have focused on plasmids. Recently, GO was covalently functionalized with (3-aminopropyl) triethoxysilane (APTES)/spermine and hydroxyapatite (HAp) to deliver gene p53 and with Survivin–green fluorescent protein–HSV-TK plasmid, respectively. These plasmids can induce the expression of specific genes, such as P53, which is a tumor suppressor. HSV-TK is a herpes simplex virus thymidine kinase gene, also known as the suicide gene. The results have shown that the functionalization APTES/Spermine and HAp completely retarded the migration of these plasmids, with 50- and 1.5-fold gene expression increases, respectively, leading to cancer cell death by apoptosis [14,31].

As presented in Table 2, regardless of the payload type, GO-based carriers modified by combined agents showed a carrier/payload ratio of less than 5, indicating that the presence of more positive charges (i.e., amine groups) on the GO surface led to more efficient loading and the protection of genes. Although GO flakes can easily penetrate cancer cells, the cellular internalization was enhanced by adding CPPs, providing more positive charges for gene derivatives to be absorbed.

On the other hand, it was shown that NAS-24 and cytochrome C aptamers were loaded directly on GO through π-π stacking interactions in the absence of mediators. The interaction between the NAS-24 aptamer and vimentin, an intermediate filament protein overexpressed in cancer cells, can cause cell apoptosis, which can be detected by the fluorescence-labeled cytochrome C aptamer. Immobilizing both aptamers on bare GO could completely prevent aptamer migration due to strong π-π interactions. This implies that it is possible to fabricate GO carriers without polymer modifications, which can also be as effective as functionalized GO [72].

## 3. Toxicity

A comprehensive analysis of the in vitro and in vivo cytotoxicity and biocompatibility of graphene oxide is an essential aspect to consider for the development of nanoparticles for drug and gene delivery. Graphene-based materials can induce cytotoxicity through a few different mechanisms: (1) interactions with the lipid tails of cell membranes, which induce the extraction of hydrophobic cholesterol and the creation of pores that damage the integrity of the membrane; (2) direct physical contact interactions of the sharp edges of GO flakes with cells, causing the rupture of the plasma membrane [112]; (3) penetration inside the cells and stimulation of excessive production of ROS, which cause dysfunction at the mitochondrial level; (4) the release of lactate dehydrogenase (LDH), which causes damage to the cell membrane [113,114]; (5) intranuclear penetration and an interaction with DNA, which induce genotoxic effects [115]; (6) the induction of an immune or inflammatory response, which activates apoptotic pathways (IKK/IκBα/NF-κB and BAX/BCL-2) [116].

Numerous studies have analyzed characteristics of GO to determine its potential use for biomedical applications [69,117,118]. These studies have shown that the toxicity depends on the complex interactions of several intrinsic characteristics of GO and extrinsic factors. The physiochemical properties, such as the shape, size, oxidation, and functionalization [69,119,120], are some of the main characteristics of GOs that can influence their interaction with the biological system. In addition, other experimental parameters, such as the dose of administration, time of exposure of GOs to cells, and type of cell line, represent important discriminating factors that can result in variable toxicity responses [118,119,121]. Therefore, to evaluate the toxicity of GO carriers, it is necessary to consider all of these parameters and to find the right ranges and combinations that allow the minimum cytotoxic response to be obtained on normal cells and the maximum one to be obtained on cancerous cells for breast cancer detection and elimination. Although the GO toxicity can be beneficial for the death of cancer cells, it raises the demand for the control of GO’s functionalization to avoid the death of non-specific cells in targeted drug and gene delivery.

### 3.1. In Vitro Toxicity

In vitro cell viability tests are the common steps taken to study and verify the influence of various intrinsic and extrinsic parameters on the material itself in order to understand its cytotoxicity, before continuing to more complex and expensive tests performed in vivo [121]. The structure, morphology, and size of GOs are the major parameters that can influence their interaction with cells. Through in vitro experiments, it is also possible to expose specific concentrations of GOs to different cell lines for defined periods of time and to test the influence of the time and concentration on the cytotoxicity in different cell types [118,122]. Following the exposure of cells to GOs, internalization studies highlight the mechanisms of cellular uptake and intracellular distribution that are influenced by the surface charge of the nanocarriers and their exposed functional groups [114,117]. The observation of the nanoparticle’s journey inside cells allows an understanding of the fate following payload release. It shows where the GO carriers accumulate and whether they are biodegraded within specific intracellular organelles (e.g., endosomes, lysosomes) or excreted by exocytosis and captured by phagocytic cells (i.e., macrophages) [123].

#### 3.1.1. Dose-, Time-, and Cell-Line-Dependent Cytotoxicity

GO is known to induce a dose- and time-dependent cytotoxicity [124,125]. Yue et al. [117] demonstrated that naked GO exposed to MCF-7 cells could induce dose-dependent cytotoxicity in the range of 0–20 μg/mL, decreasing the cell viability by about 20% within 48 h. However, these results proved the non-toxicity of GO in these concentrations. The same trend was reported by Ribeiro et al. [114], in which MCF-7 cells showed a decrease of about 12% in viability when exposed for 72 h to 48 μg/cm^2^ GO. However, other studies found a different behavior of GO on breast cancer cells. Alibolandi et al. [90] showed that bare GO showed high cytotoxicity after 24 and 48 h, even at low concentrations (<70% cell viability with concentrations > 10 μg/mL and 20 μg/mL in 4T1 and MCF-7 cells, respectively). 

Regarding the time-dependency of the cytotoxicity, the literature has reported conflicting results. Some studies did not find any relationship between the exposure time to GO and the decrease in viability [90,114], while others observed time-dependent cell death [118,119,122]. Chowdhury et al. [122] stated that GO nanoribbons decorated with amphiphilic lipid 1,2-distearoyl-sn-glycero-3-phosphoethanolamine-PEG (DSPE-PEG) could cause a time- and dose-dependent cytotoxicity, even though the cell viability measured in MCF-7 cells for the highest concentration tested (i.e., 400 μg/mL) showed only decreases of 15% and 20% after 24 and 48 h treatments, respectively. Thus, there is no clear agreement regarding the influence of the concentration and exposure time of GO on its cytotoxicity to breast cells. In addition, Chatterjee et al. [126] showed that GO could cause dose-dependent cytotoxicity and apoptosis due to damage and a consequent loss of structural integrity of the plasma membrane. This phenomenon can be explained by the strong physical interaction that forms between GO and the phospholipid bilayer. 

Some studies have investigated the possibility of using rGO, which is characterized by a higher hydrophobicity that leads to a higher affinity for the cell membrane [127]. In the work by Krętowski et al. [118], the cytotoxicity of rGO nanoplatelets was analyzed on different breast cancer cell lines (T-47D, MCF-7, MDA-MB-231, ZR-75, Hs 578T) through LDH and propidium iodide staining (PI) tests. The cells were incubated with different amounts of rGO, ranging between 25 and 300 μg/mL, and with two different exposition times (24 and 48 h). The results showed that rGO can cause an increased time- and dose-dependent cytotoxicity with respect to MDA-MB-231 and ZR-75-1 breast cell lines. The results showed that after 48 h of exposure to 50 μg/mL of rGO, 30% of the cultured MDA-MB-231 cells and 50% of ZR-75-1 cells were apoptotic, while 8% of the former and 2% of the latter were necrotic. Equivalent results were obtained in the study by Farell et al. [119], in which lipid–rGO-nanocarrier-treated MDA-MB-231 cells showed an increase in cytotoxicity by about 3-fold with respect to the controls when exposed to a lower concentration (i.e., 10 µg/mL). The results were not the same for MCF-10A and MCF-7 cells. For these cell lines, there was no variation observed in the cell death level during the treatment period with increased concentrations (i.e., 10, 50 and 100 µg/mL) of bare and lipid–rGO compared to the control samples. These results proved that rGO is more cytotoxic than GO, demonstrating a strong interaction between the graphene and cells within a limited period of time. Since the toxicity depends on the physicochemical properties of the materials, the use of reducing agents to deoxygenate GO can lead to the formation of materials with different characteristics (i.e., particles size, density of functional groups on the surface) and different toxicity levels. In fact, following some reduction treatments, rGO has been shown to aggregate due to van der Waals interactions and has exhibited different sizes. This characteristic is known to be a major cause of increased toxicity, as will be discussed in the following sections.

More clarification is needed to better understand in which conditions dose and time influence the toxicity of the GO carriers. The differences between the results found in the literature can be attributed to the different physicochemical properties of graphene-based nanomaterials, such as the surface functional groups, the shape and size of the flakes, and the unique characteristics of the treated cell lines (Table 3). For instance, MCF-7 cells are not affected by the dose or time of exposure to GO with respect to other breast cell lines (i.e., MDA-MB-231, 4T1, ZR-75-1) [90,114,118,119]. The lower sensitivity of this cell line to GO can be caused by its different responsive expression levels of apoptotic and autophagic genes when exposed to the material. Different cell lines are characterized by distinctive characteristics, such as their morphology, dimensions, apoptotic genes, cell cycle activity, and expression of estrogen receptors [118]. Nonetheless, Chowdhury et al. [122] tested the dose- and time-dependent cytotoxicity of GO on all studied cell lines, including MCF-7 cells, while in other studies MCF-7 cells demonstrated only dose-dependent cytotoxicity [90,114].

Even though the dosage and time were proven to highly influence the cytotoxicity of GO in several studies, the dependency was not found in others (Table 3). Instead, it was noticed that other parameters can be modulated, allowing decreased cytotoxicity in similar experimental conditions, such as by modifying the biocompatible functionalization and size of the GO [113,114,128].

#### 3.1.2. Charge- and Functionalization-Dependent Cytotoxicity

Cellular interactions are profoundly affected by the charge and chemical composition of GO, having a high negative charge density due to its numerous oxygenated functional groups. While it is well known that the cellular membrane has a net negative or neutral electrical charge, the negative zeta potential of GO measured in different studies (Table 4) reduces the cytotoxicity due to the same charge of the cellular membrane that reduces their interactions [133]. However, electrostatic interactions between GO and membrane lipids may arise due to the presence of some positively charged lipids on the plasma membrane. In addition, hydrophobic interactions between negatively charged GO and lipids can lead to GO adsorption and membrane damage [134]. Nevertheless, even if GO was shown to induce a certain degree of cytotoxicity, the possibility of modifying its surface with biocompatible polymers (e.g., PLA, PLGA, PAMAM) could significantly decrease its toxicity, both in vivo and in vitro [135]. Moreover, it has been seen that GO tends to aggregate in cell culture medium and biological fluids, due to the presence of serum [34]. The formation of GO aggregates is detrimental to its function, as once the aggregates are formed, the GO carriers will no longer have the same initial structural characteristics, reducing their ability to penetrate the cells and to release the payload. Furthermore, aggregates of GO carriers could have large dimensions, making them unable to cross biological barriers and accumulate inside cells and tissues, increasing the toxic effect [57]. Thus, GO in physiological solutions induces unfavorable reactions for putative applications in the biomedical field, limiting its use without further surface modifications [92]. As previously said in Section 2, studies have demonstrated that the use of polymers, such as PEG [86], PEI [92], PVA [93], DEX [90], PAA [94], and chitosan [95], which were covalently bonded to GO functional groups, can enhance the biocompatibility and stability of GO-based drug or gene delivery systems. However, care must be taken, as many of the polymers that are often used are cationic and can increase the zeta potential of the carriers to positive values (see Table 4). Excessively high positive charges can induce higher cytotoxicity due to the strong physical interactions with the cell membrane, which can casue its damage and breakage [64].

GO can induce toxicity due to its different properties; however, the toxic effects are also likely to be highly dependent on its accumulation in certain tissues and the time of exposure to cells. Therefore, the functionalization of the nanocarriers with targeting molecules (e.g., antibodies specific to certain cells) is the main approach to control the accumulation and exposure time of the nanomaterials to specific sites. Targeting molecules, which can specifically recognize the tumoral cells, allow nanoparticles to accumulate in the cancer site, inducing the cytotoxic effect mainly only in the circumscribed area of the tumor [69]. In addition, functionalization can increase the specificity and recognition potential of the nanocarriers from the body [137].

#### 3.1.3. Size-Dependent Cytotoxicity

GO can induce cell death via various pathways. In general, the greater toxicity of small flakes is recognized [112,138], even if other studies have not found this correspondence [117]. Different sizes can induce different cytotoxicity effects (Figure 4). The toxicity pathways induced by GO can be of various types: (1) chemical induction of intracellular stress (ROS production) [122,138]; (2) mechanical induction by the physical interaction of GO with cell membranes, which can lead to their damage and breakage [112]; (3) genetic induction through the interaction with cellular DNA in the nucleus and its fragmentation or production of chromosomal aberrations [138,139]. The first pathway is observed in several studies [122]. Small GO flakes (i.e., ~10 nm) have been shown to induce increased oxidative stress [138]. The cause could be found in the procedure of breaking GO flakes to produce smaller ones. During this process, more edge defects are inserted, which can turn into active sites for ROS production [112]. The influence of the GO thickness on ROS production was also investigated in one study [138]. Single-layer and 4-layer GOs were assessed on MCF-7 cells. Again, smaller dimensions (i.e., single layer GO) induced greater oxidative stress with consequent greater cell death. Although this phenomenon seems to have been recently recognized as the main effect of nanomaterial-induced toxicity, other mechanisms seem to appear as the size of the GO flakes varies. As the size decreases, the nanoparticles present a lower surface-to-perimeter ratio, thereby exposing more of their irregular edges to cells.

This phenomenon could induce the second pathway of cytotoxicity. The interaction of cells with the sharp edges of flakes may cause the damage and rupture of the plasma membranes. When the size is small, the mobility of the GO is high, which may increase the probability and number of interactions that a single flake has with the cell. Akhavan et al. [112] observed this phenomenon by studying the efflux of RNA from cells. They found that GO flakes of 11 ± 4 nm were shown to induce much greater RNA efflux from cells than graphene measuring 418 ± 56 nm, for which no significant RNA efflux was recorded, except for higher concentrations (i.e., 100 μg/mL).

In some cases, despite inducing cell death, GO flakes are not responsible for damaging the cell membrane (i.e., no RNA efflux or significant levels of LDH have been recorded) [112,140], nor for the significant production of ROS [138]. These studies suggest the presence of other mechanisms capable of inducing necrosis. Among these mechanisms is genotoxicity. GO flakes can penetrate the nuclear membrane and interact with the cellular DNA inside the nucleus [139,141]. Following this interaction, DNA fragmentation and chromosomal aberrations can occur and induce apoptosis [138,142]. 

These considerations reveal the importance of producing and defining the size of the manufactured particles accurately. Predicting the main pathway of cellular toxicity, combined with the possibility of adequately functionalizing such GO flakes, could allow apoptosis to be driven in target cells in a highly controlled way.

#### 3.1.4. Oxidative Stress

As mentioned above, oxidative stress is a phenomenon that normally occurs inside the body [118]. An imbalance in the regulation of the production and elimination of these reactive oxygen species within cells can lead to cell mutations, such as the emergence and development of tumors [143]. They are usually characterized by their ROS production in much higher concentrations and by a limited antioxidant enzymatic activity compared to healthy cells [119]. Various compounds can be used to induce this oxidative stress, such as lipid peroxides, oxidized proteins, and sugars. These are normally compensated by the presence of reduced compounds, such as the reduced form of glutathione (GSH) [118]. It plays a fundamental role in protecting cellular activities that can be affected by the presence of free radicals.

A decrease in the antioxidant GSH and a consequent increase in the ROS level have typically been observed following the administration of GO and rGO [121]. The increase in oxidative stress can cause lipid peroxidation, DNA and protein fragmentation, and mutation due to their oxidation [121]. This process can lead to the breakage, modification, or further cross-linking of the molecular chains. Cellular proteins and DNA can, therefore, undergo modification or even a complete loss of their biological activities [144].

Two possible explanations for ROS-induced cell death following GO internalization can be found in the literature [118,145,146]. The first signaling pathway that can be activated is due to the interaction of GO with the electron transport system that induces the overexpression of H_2_O_2_ and hydroxyl radicals, resulting in the oxidation of cardiolipin and its release into the cytoplasm of the hemoprotein contained in the mitochondria. This mechanism consequently induces the release of the cytochrome C complex, which stimulates the release of Ca^2+^ from the endoplasmic reticulum. This process in turn activates the caspase cascade, resulting in cell death [146]. The other possible mechanism involved may be the GO-driven induction of mitogen-activated protein kinase (MAPK) (i.e., JNK, ERK, p38) and transforming growth factor-beta (TGF-β) signaling pathways, resulting in the activation of Bcl-2 proteins. The latter could be directly responsible for the initiation of mitochondria-induced apoptosis [145].

Therefore, it may be possible to exploit GO-based nanocarriers to selectively induce an increase in oxidative stress at the oncological site to induce cancer cell death [119]. In several studies performed on breast cancer cells (i.e., MCF-7, MDA-MB-231 and 4T1 cells) [119,135,147,148], naked and functionalized GO have demonstrated the ability to induce cell death in such pathological cells by inducing the generation of ROS species. Farell et al. showed different trends of oxidative stress in two cancerous (i.e., MCF-7 and MDA-MB-231) and one non-cancerous (i.e., MCF-10A) cell line (Figure 5) [119] when exposed to naked and lipid-functionalized rGO carriers. However, the ROS levels recorded in all cell lines in response to naked rGO administration were remarkably high. This could indicate that non-functionalized rGO is not an adequate material for breast cancer treatment, as such high ROS concentrations could result in an increased risk of mutations of normal breast cells into tumor tissue. Even when tested in lower doses than GO, rGO was shown to induce the same or higher levels of cell death, depending on the time and dose of exposure to cells. This phenomenon can be related to the higher affinity of rGO to the cell membrane, which increases the hydrophobic interactions between them. In addition, some chemical agents used for GO reduction can generate metallic impurities and organic contamination, causing alterations in the interaction with cells, leading to cell membrane damage and apoptosis. 

#### 3.1.5. GO Clearance from Cells and Biodegradation

As mentioned previously, the GO concentration is one of the parameters that plays a key role in regulating the toxicity of GO [90,118,119]. When a high dose of GO accumulates in cells, it causes their death more rapidly [114]. Therefore, a particularly important aspect to consider is the fate of GO flakes following the delivery of therapeutic agents. Most of the studies in the literature investigate the efficiency of GO in loading, transporting, and delivering payloads specifically into pathological cells, and few studies so far have focused on the disposal of such nanocarriers once they have accomplished their task. This aspect is critical to understand the efficiency of GO as a transporter for gene therapy. It must not exhibit toxicity on its way to the target site in its complete laboratory-prepared form, but it must also be completely disposable without causing toxicity once disassembled [123]. As explained in Section 2, it is usually decorated with other molecules, typically PEG or cationic polymers, to increase its stability and half-life within the bloodstream before being recognized by macrophages during the cargo delivery [34,86]. Once the payload is administered, the separation of the GO from the polymer chains may allow it to be more easily recognized by phagocytic cells and degraded.

The degradation of GO can be catalyzed by peroxidase enzymes, which are naturally occurring in plants and humans [149,150,151]. They are responsible for the oxidative biodegradation of graphene driven in living beings by the presence of reactive intermediates formed during their catalytic cycloadditions, such as plant horseradish peroxidase (HRP) [150], inflammatory human myeloperoxidase (hMPO) [149], and eosinophil peroxidase [151]. These reactive species can convert halides into strong oxidants (i.e., hypochlorous acid, HOCl) that can induce GO biodegradation [152]. HOCl produced by peroxidase from H_2_O_2_ could be a major cause of the biodegradation of carbon nanomaterials [149,150,151]. Several studies have confirmed that recombinant hMPO is able to degrade single-layer and multi-layer GOs [153,154], and that the degradation products are non-toxic [155]. Through other studies, researchers have shown that graphene-based materials can be degraded by hMPO [153], recombinant eosinophil peroxidase [156], and HRP [153,157].

Macrophages activated by the presence of GO begin to secrete MPO, which converts H_2_O_2_ and chloride ions into HOCl. Notably, during this oxidative process, the epoxide groups of GO were transformed into carbonyl groups, causing C-C bond breakage and the fragmentation of GO into sp^2^ aromatic domains [158]. Therefore, macrophages can degrade GO flakes after their internalization [123]. As small GO fragments (i.e., a few tens of nm) have been shown to exhibit photoluminescence properties [159], these properties can be exploited to investigate the effective disassembly and degradation of GO-based nanocarriers [123]. In the study by Kim et al. [123], the photoluminescence of GO, both naked and functionalized with PEG and PEI, was measured after treatment with HRP and H_2_O_2_ to simulate the intracellular environment of macrophages. The intensity was increased, indicating that fragmentation of the GO had occurred. Moreover, even if it has been reported that the cell internalization of GO is size-dependent in different cell lines [52,53], different studies have proven that macrophages (i.e., PMƟ and J774A.1) are not always affected by the size and dosage during GO uptake [117]. All of these findings may demonstrate the potential for naked graphene-based materials to be degraded within phagocytic cells after the payload delivery [69,123].

On the other hand, micro-sized GO flakes (i.e., lateral dimension < 1 μm) have been shown to be internalized more readily by mammalian cells through endocytic mechanisms (i.e., clathrin-dependent, caveolin-dependent, or clathrin- and caveolin-independent endocytosis; phagocytosis; micropinocytosis) compared to larger GO [24,34,160]. All of these mechanisms are due to the fact that once internalized, GOs were found to be accumulated and confined within intracellular vesicles (i.e., endosomes, lysosomes, phagosomes, and macropinosomes) [161]. Inside lysosomes, GO can be degraded through acid hydrolases, which include proteases, lipases, nucleases, glycosidases, phosphatases, and sulfolipases [24,160]. Several studies have simulated the co-localization of GO flakes within lysosomes, even after payload release [54,117,160,162]. 

### 3.2. In Vivo Toxicity

In general, the in vivo fate of nanomaterials is influenced by several factors. In addition to the characteristics and parameters we have discussed in the previous section, there are also the routes of administration, the physiological environment that is tested, the interactions with the blood, the proteins and immune system of the host, and the presence of many different barriers [163]. 

The protein corona is one of the first factors to be addressed, which causes the rapid change of GO in the bloodstream [164]. This protein coating induces the size change of GO that substantially influences the interactions with the cells [34]. These interactions include the internalization, biodegradation, biodistribution, and delivery to target sites. Although preliminary in vitro studies are essential to set the basic parameters and perform a first screening of the tested materials, in vivo studies are equally essential to fully understand the behavior of the designed systems in living organisms. In fact, results obtained in vitro are not always reproducible within a biological system, due to several peculiar responses that cannot be predicted or simulated, such as the inflammatory response, the biodistribution, and the clearance from the body.

#### 3.2.1. Inflammatory Response

The body’s non-specific immune defense relies on phagocytosis of foreign molecules via macrophages, providing a significant barrier to intravenous injections of GO nanocarriers. In addition, GO nanocarriers have a high probability of being removed by macrophages before reaching their destination and may initiate an inflammatory reaction [24,164]. In particular, when in the bloodstream, larger nanoparticles (>200 nm) have a higher risk of being recognized and sequestered [44,53,123,137] by the phagocytosing cells of the immune system (i.e., macrophages, dendritic cells, neutrophils, and B lymphocytes), which are responsible for recognizing the foreigner and destroying it via enzymatic digestion, as explained previously [31]. When macrophages are activated by invader molecules, they start to release cytokines and chemokines, resulting in the accumulation of neutrophils and monocytes. Like all foreign materials, GO could also cause the secretion of cytokines, such as IL-1α, IL-6, IL-10, and tumor necrosis factor-alpha (TNF-α), as well as chemokines, including monocyte chemoattractant protein-1 (MCP-1), macrophage inflammatory protein-1α (MIP-1α), and MIP-1β. This leads to cell internalization and removal by macrophages via phagocytosis [165,166].

In the study by Yue et al. [117], GO flakes with lateral sizes of 2 μm and 350 nm were brought into contact with phagocytic and non-phagocytic cells and their responses were compared. The results showed that macrophages have a higher ability to internalize GOs than non-phagocytic cells. The motivation might lie in the fact that macrophages are able to overcome the strong electrostatic repulsions that are generated between the negatively charged surfaces of the GO and the cell membrane. This recognition may be mediated by the phagocytosis receptor Fcγ [117]. As previously mentioned, it has also been shown that GO flakes with lateral sizes from nanometers to microns can be similarly internalized by macrophages. However, GO (with micron size) activates stronger inflammatory responses, characterized by significant increases in cytokine levels (i.e., IL-6, IL-12, TNF-α, MCP-1, and IFN-γ) [117]. One possible explanation for this size-dependent phenomenon is the strong steric effects caused by larger GO when it encounters cells [117]. This result has also been proven in other studies, in which larger GO flakes have been revealed to induce increased cytokine and chemokine production and immune cell recall due to increased interaction with phagocytic cell membrane receptors, which induce the activation of the immune response via the NF-kB pathway. This mechanism was demonstrated to induce programmed cell death in MCF-7 breast cancer cells [167].

It can be concluded that GO flakes with lateral sizes at the nanoscale may induce a decreased inflammatory response, resulting in a more efficient solution to deliver therapeutic agents to the human body. For examples, some studies [168,169] have shown that GO recognition by macrophages could be kept under control by injecting GO flakes with a size smaller than about 150 nm. Another important aspect to avoid recognition by phagocytic cells is functionalization with hydrophilic molecules, which weaken the opsonin–protein interaction. Since the recognition and elimination of nanoparticles by phagocytes is mediated by opsonization (i.e., adsorption of immunoglobulins, serum proteins, and complement proteins on the surfaces of nanoparticles, causing the formation of the protein corona), the inhibition of opsonization is generally taken as a possible solution [170]. A highly utilized approach is PEGylation [171], or functionalization of the nanocarriers with target molecules. Hydrophilic PEG generates a shield through its extended molecules, inducing a repulsive barrier between the GO and circulating proteins, while targeting molecules increase the specificity and recognition potential of the NPs from the body [137]. However, making GO smaller than 200 nm in size would be preferred to limit their recognition by immune cells when administered intravenously and by the inflammatory response [17,44,123].

#### 3.2.2. GO Biodistribution and Clearance from the Body

Another fundamental aspect to determine the effective toxicity of graphene-based nanomaterials is their biodistribution and excretion from the biological environment as assessed via in vivo measurements. Generally, GO was found to distribute in tissues and organs, inducing different response to cells cultured in vitro. In one study, PEG-enhanced functionalized monolayer carbon nanotubes were found to be non-toxic when injected in vivo into animal models (i.e., rats and rabbits), although the results showed some level of toxicity when used in in vitro models [163]. This result may indicate that the toxicity of a material could be concentration- and time-dependent, as analyzed in Section 3.1. In fact, for concentrations of nanomaterials localized in cultured cells, a certain toxicity can be found, which on the contrary was not detected when carriers were injected into an organism where they were dynamically transported in the blood and continuously circulated in different tissues and organs [163].

However, it has been confirmed by several studies that GO injected intravenously tends to accumulate preferentially in some tissues and organs [166,172]. Through in vivo studies, it has been shown that most intravenously administered nanoparticles tend to accumulate in the liver and spleen [173,174]. Nevertheless, these studies also showed that toxicity was not observed, as predicted from the in vitro studies (Figure 6) [64,166]. Moreover, only a small fraction of the total number of injected nanocarriers were demonstrated to accumulate at the target pathological site (i.e., approximately 1−10%) [172].

The size plays an essential role in the journey of GO in in vivo tests. After the injection, the circulating GO nanoparticles start to diffuse through tissues and accumulate in organs. In the same report, Zhang et al. [76] investigated the biodistribution of intravenously administered GO-based particles in mice, showing that GO usually accumulates preferentially in organs, such as the liver, spleen, lungs, and bone marrow. As explained in the previous section, biological barriers are present inside the body, which can limit the circulation of GO within the body. Sometimes it is also possible for GO to pass through them and accumulate in different compartments. One study illustrated that GO particles with a size of 54.9 ± 23.1 nm were unlikely to cross the blood–testicular and hemato-epidymal barriers after intra-abdominal injection. Interestingly, further tests showed that the sperm of mice analyzed after GO treatment was unaffected, even when administered at a high dose (i.e., 300 mg/kg) [175]. Other barriers such as placental and blood–brain barriers were also evaluated. In both cases, nanoscale GOs were able to pass through these membranes and diffuse into the compartment, affecting the fetus and the nervous system, respectively [176]. These observations highlight the need to control the characteristics of graphene-based particles in such a way that efficient targeting of the disease site will be achieved, with the aim of minimizing their interaction with healthy tissues as much as possible.

In general, when a non-degradable nanomaterial is too large to be filtered out of the body and then excreted through the kidneys, it accumulates in the tissues. The estimated time spent inside them is about 8 months [163]. GO-based nanomaterials are usually considered to be poorly degradable in vivo. Therefore, their functionalization by adding hydrophilic functional groups on the surface represents a possible strategy to optimize their biocompatibility and degradability in vivo. However, there are no prolonged studies (i.e., over 1 year) regarding the distribution and maintenance of GO within the body. This aspect raises the need for further in vivo analyses to better understand its toxicity over time.

In addition to size, the surface functionalization of graphene flakes via biocompatible polymers also influences their biodistribution in vivo. For example, GO modified with FA or heparin could increase the ability of nanocarriers to target and penetrate 4T1 breast cancer cells in vivo by recognizing FA receptors or receptors for advanced glycation end products (RAGE), respectively [177].

In the study by Yang et al. [178], PEG modified GO-based nanoparticles with a monoclonal antibody modification were tested against follicle-stimulating hormone receptor (FSHR) to specifically target metastatic nodules of MDA-MB-231 breast cancer cells within the lungs. The metastatic tumor targeting the efficiency of GO conjugates was investigated in a mouse model. The obtained results showed that GO nanomaterials have excellent stability and high specificity for FSHR, which was demonstrated by the rapid tumor uptake of the GO conjugates. The accumulation of these GO-based materials was very low in healthy tissues, and their elimination from the bloodstream was much faster than that detected from tumor nodules. These results demonstrated the efficiency of such nanocarriers to specifically target the tumor site and accumulate in it (Figure 7A).

In the study by Chen et al. [64], GO was functionalized with chitosan (GC) and the EpCAM antibody (GCE) as a targeting agent for delivering Survivin-siRNA in MCF-7 in vitro and in vivo. The targeting effect of the antibody and the biodistribution in vivo were analyzed in nude mice, as a model of a human breast cancer xenograft. The test was conducted by injecting GO nanocarriers intravenously followed by in vivo fluorescence imaging after 1, 2, 4, and 8 h. The results showed that the fluorescence intensity of functionalized GO was higher than the control, demonstrating the specific targeting effect of the EpCAM antibody on MCF-7 cells. In addition, through the dissection of the main organs (i.e., heart, lungs, livers, kidneys, and spleen), it was observed that a higher accumulation occurred in kidney and liver (Figure 7B). Moreover, comparing the nanoparticles functionalized with the antibody (GCE/siRNA) and a non-targeted control (GC/siRNA), it was evidenced that the targeting agent was effective in increasing the accumulation potency of the nanocarriers (Figure 7C) [178].

Regarding the excretion of GO from an organism, there are several possible pathways, depending on the organ in which the particles are located. For example, the nanoparticles accumulated in the lungs have shown greater difficulty in being eliminated. Furthermore, they have often been shown to cause cellular infiltration, inflammation, and the onset of granulomas and pulmonary edema with greater ease [179]. In the liver, GO-based nanomaterials can be eliminated via the hepatobiliary pathway via the duodenal bile duct [172]. It was pointed out that GO was able to overcome these barriers and be excreted even with larger flakes. This is probably due to its ability to fold [180]. However, large GO particles (i.e., >200 nm) usually appear to accumulate via physical splenic filtration, as well as being prone to being cleared more rapidly from the circulation by macrophages [172], as explained in the GO biodegradation section. Conversely, small GOs, with dimensions under the cutoff for renal filtration (i.e., about 5 nm), can enter the renal tubules to be rapidly excreted without any toxicity via the urine [24]. Yang et al. [178] observed that when injected intravenously, a high fraction of GO particles was excreted through the hepatobiliary route, as the size of the nanomaterials was greater than 5 nm. Some studies have also observed that the urinary elimination of GO was size-independent and influenced its excretion rate. In order to confirm this hypothesis, large GO flakes (1−35 μm) were tested and were excreted more slowly than small (30 nm–1.9 μm) and ultra-small (10−500 nm) GOs [181]. 

The elimination pathways of GO in vivo are not yet clear and defined. Nevertheless, the renal and fecal pathways appear to be the main pathways of excretion. However, the results published to date have shown conflicting results regarding the biodistribution and excretion of GO from organisms in vivo [172].

## 4. Controlled Release Strategies

The controlled release of drugs and genes from delivery nanocarriers is a mechanism that has been extensively studied in recent years. One of the main limitations to overcome is the non-specific treatment that results from the uncontrolled release of the payload inside the body during the circulation of the nanomaterials in the bloodstream [88]. The goal to be achieved is the specific accumulation of the nanoparticles in the tumor site and the release of the drug/gene only once internalized within the pathological cells [182]. This would allow an increase in treatment efficiency, as a high percentage of nanocarriers would transport the therapeutic agent directly to the oncological sites [123]. This mechanism would lead to a decrease in the dose of nanocarriers necessary to administer, with consequent advantages of low toxicity and limited side effects [53,168].

Several approaches have been devised to achieve payload release in a controlled manner. Some of these mechanisms exploit factors intrinsic to the human body, such as changes in pH [65,95,162], temperature [183,184], or the presence of reducing [123,185,186] or enzyme-rich [186] environments. Other mechanisms can instead be induced externally to guide and control the behavior of the injected nanoparticles once the target site is reached. Some examples of these extrinsic mechanisms are the administration of near-infrared radiation [53,123,136] to induce the photothermal effect on the GO and the controlled injection of specific molecules able to compete with the payload for the bonds on the surface of GO [187,188]. Table 5 summarizes the primary mechanisms used to date to induce a controlled release of the therapeutic agent from graphene-based nanoparticles. Some studies have presented the use of these mechanisms individually, while others have also proposed the possibility of fabricating multimodal carriers capable of responding to several different triggers to maximize the transport and release efficiency. 

### 4.1. pH-Sensitive Platforms for Control Release

In Section 2.2, we described that GO has a unique surface structure due to its large number of hydrophilic and pH-sensitive groups, which confer a highly anionic charge given the negative surface charges deriving from their ionization. It is well known that the tumor environment has a higher acidity level, as tumor cells have a high rate of glycolysis and excrete lactic acid [91,199], lowering the pH (pH = 5.7–7.8) compared to the physiological level (pH = 7.4) [162,182]. In addition, some intracellular compartments also exhibit an acidic surrounding, which can be exploited to release the payload only when the nanoparticles are internalized into the specific cancer cells. Late endosomes and lysosomes show pH values of around 4.5–5.5, in contrast to the pH of the cytosol, which is about 7 [199]. This pH gradient is an excellent trigger that can be used to stabilize the payload at physiological pH and to perform the selective release when the nanocarriers are tuned to a pH environment below a certain trigger threshold [97].

Many experiments have been carried out by exploiting GO and pH for smart drug delivery. The efficient loading of various anticancer drugs onto GOs via π-π stacking and hydrophobic interactions and their pH-controlled release has been widely demonstrated (Table 6). Many studies have assessed the efficiency of this mechanism when hydrophobic drugs are to be delivered in hydrophilic body fluids [88]. Under basic pH conditions, such drugs are usually deprotonated and can be readily loaded onto GO flakes, which are also hydrophobic under the same neutral pH conditions. Conversely, in an acidic environment, such drugs protonate and become more hydrophilic and more soluble in water and less attracted to GO, facilitating their release from the nanoparticle surface [88,200]. DOX is one of the most widely used drugs that has been used to treat several types of cancer, particularly against breast cancer [18]. At low pH, it becomes more hydrophilic due to the protonation of the daunosamine group, which weakens the hydrophobic π-π stacking interaction, causing the release from GO-based nanoparticles [88].

To stabilize genes on GO, pH-sensitive bonds can be a way to exploit this trigger in gene therapy. Such pH-cleavable covalent bonds can be formed between the cationic molecules and the GO, thanks to its large surface rich in reactive functional groups. Once it reaches the tumor or intracellular site, the bond is broken, leading to the disassembly of the nanoparticles and the release of the gene. Some of the most important pH-cleavable bonds are shown in Figure 8, which were used in the production of nanocarriers for the transport of pH-sensitive genes [97]. The amide bond, formed between the amine group (−NH_2_) of cationic molecules and the carboxylic group (−COOH) of GO, is the most widely used acid-sensitive linker in GO-based systems for the delivery of genes against tumors [17,64,98]. 

Chen et al. [64] and Xu et al. [162] explained the functionalization of GO with chitosan through amide bonds and found that the release efficiency of siRNA is significantly higher at acidic pH (5.0) compared to physiological pH (7.4) in breast cancer cells. However, the majority of studies in the literature have reported a low gene release efficiency when using chitosan. This could attribute to the further protonation of chitosan, which may increase the positive charge of the GO-based carriers, which would increase the electrostatic binding force to the transported genes [201]. In these studies, it has been shown that the pH response was given by the pH-sensitive molecule attached to GO instead of by GO itself. Hence, when the nanocarrier disassembles in acidic pH, the efficient delivery of the therapeutic agent is due to the ability of the functional molecules to release genes. Izadi et al. [65] functionalized carboxylate GO with chitosan and hyaluronic acid to transport HIF-1α-siRNA (i.e., Hypoxia Inducible Factor-1) and Dinaciclib. The aim was to silence HIF-1α, a factor that influences the spread of the tumor by regulating the expression of genes involved in cellular growth and in the blockade of cyclin-dependent kinases (CDKs) in CD44-expressing cells, including 4T1 breast cancer cells. The results showed a non-significant difference in the gene release in neutral and acidic pH values. Similar observations were made in other studies on different cells lines [182,201], demonstrating the presence of contradiction in the efficiency of chitosan in its pH-sensitive functionalization for gene delivery. Similar conclusions were observed by Yadav et al. [17], who decorated GO flakes with PEG and the cationic dendrimer PAMAM to deliver siRNA in MDA-MB-231 cells. The release efficiency of the nanocarrier was assessed after 72 h of incubation at pH 6.5, 7.0, 7.5, and 8.0, obtaining results of 11.8, 38.9, 60.9, and 38.4%, respectively. Therefore, it was seen that at pH 7.5–8, the percentage of primary amines was lower, while at pH = 6–7, the quantity of ammonium ions increased, generating a higher net positive charge of the dendrimer. This phenomenon led to a more strongly complexed oligonucleotide due to the greater electrostatic interactions that were formed. This resulted in a slower release rate, which can interfere with the processing of genes within the cell and their transcription [201].

It can be noted that in the literature, there are some contradictions about the efficiency of the pH-stimulated gene release using cationic molecules as an intermediate for the functionalization between GO and nucleotides (Table 6). Further studies may be necessary to better understand this modification approach, as exploiting the pH gradients naturally presented in biological and tumor environments and the ability of the GO to respond to them could be a potentially advantageous approach.

### 4.2. Reducible Intracellular Environment

Tumor intracellular microenvironments are characterized by unique features compared to non-pathological tissues. In addition to their higher acidity, cancer cells also present a highly reducing environment due to their high intracellular concentration of glutathione [204]. GSH is the most abundant antioxidant in the cells, playing a major role in regulating intracellular oxidative stress by buffering ROS, as explained in the previous section [118]. It is synthesized from the amino acids glutamate, cysteine, and glycine through the controlled action of intracellular enzymes. Nonetheless, it was observed that the expression of these enzymes that drive the production of the three GSH precursors was upregulated within cancer cells [205]. The level of GSH was higher in cancer cells than in normal cells (i.e., 10 μM and 10 mM, respectively) [206]. Therefore, the presence of GSH and its activity in terms of redox buffering had a strong influence on tumor development.

Several studies showed the selective release of drug/gene within the tumor’s intracellular environment by using GO-based nanocarriers. Indeed, it is known that the disulfide chemical bond (S-S) is sensitive to reducing environments. When molecules containing this bond are close to reducing species, they undergo cleavage [207]. On the other hand, GO offers a large surface area that is capable of assembling numerous molecules containing disulfide groups and that respond to the tumor environment for controlled payload release. We have not found specific studies in the literature that have exploited this mechanism in graphene-based nanoparticles to transport genes in breast cancer cells. For example, one study [186] designed a GO nanocarrier functionalized with bovine serum albumin (BSA) in the presence of gelatin to transport DOX into MCF-7 cells and release it in a controlled manner via pH, redox, and enzymatic treatment mechanisms. To assess the redox sensitivity of such nanoparticles, the drug release triggered by the presence of 10 mM GSH was evaluated and compared to the release obtained in its absence. Each BSA molecule has 17 disulfide bond pairs, which demonstrate high sensitivity to the intracellular tumor-reducing environment. Specifically, in the presence of GSH, DOX release was faster and 1.6-fold greater than in the absence of GSH, achieving the release of 44.3% of the total drug loaded. 

In addition, Kim et al. [123] made GO nanocarriers decorated with PEG-NH_2_ and PEI via intermediate disulfide bonding using the cysteine molecule. Such a system was used to load and transport pDNA, as it can form electrostatic interactions with the positively charged amine groups of the polymer chains. The ability to release the gene via disulfide bond degradation once internalized into tumor cells was preliminarily verified by incubation in DTT (i.e., a small molecule antioxidant) in order to simulate the intracellular reducing environment [207]. The results showed that after treatment in DTT, the disulfide bonds between the polymer chains and GO were broken, resulting in the release of pDNA. As a control, nanoparticles of GO/PEG/PEI linked via amide bonds were also fabricated and incubated in DTT. No significant release of pDNA was observed in the systems.

Other studies in the literature have exploited this mechanism to control the payload release, but most of them have focused on drug transport [186,208] or have used different nanoparticles [207,209]. Nevertheless, there is evidence that once the GO is functionalized by molecules containing disulfide bonds, oligonucleotide chains can also be loaded onto such systems. Therefore, further studies could be conducted in this field to increase the efficiency of gene transfection in breast cancer treatment.

### 4.3. Enzyme-Induced Tumor Initiation

Earlier reports and findings support the theory that the enzymatic action of extracellular proteases, such as matrix metalloproteinases (MMPs), mediates various physiological processes and signaling pathways in the intra- and extracellular microenvironments during tumor progression [199]. By regulating various changes in the intracellular biochemical factors, these enzymes play a key role in tumor initiation and development. Therefore, the tumor microenvironment shows an overpressure of these enzymes with respect to physiological tissues [186]. For this reason, the creation of specific bonds that can be recognized by the metalloproteinases between the payload and the nanocarriers could be used to achieve an enzymatically controlled release in the tumor site. To our knowledge, there are no studies that exploit this mechanism for the transport of genes. However, several attempts can be found in the literature on GO-based nanocarriers for drug delivery.

Wu et al. [186] developed biodegradable multimodal carriers with a pH-, redox-, and enzyme-triggered drug release capacity, using GO modified with bovine serum albumin (approximately 5 nm) and gelatin (GGB) in MCF-7 breast cancer cells. The results showed that gelatin, being extremely sensitive to MMP-2, undergoes degradation, promoting DOX release and cleaving BSA-DOX complexes into small sizes. Thus, it increased the amount of drug released and enhanced its penetration efficiency within the tumor cells. As a result, after enzymatic pretreatment, the cumulative DOX released from GGBD nanoparticles reached 36.4%, which was 1.8-fold higher than the release obtained without enzyme treatment.

Other studies have evaluated the drug release and transport upon bond cleavage by enzymes [78,194,210], which was carried out on different cell lines. When it is not possible to attach the drug directly to GO, specific linkages can be formed using peptides or polysaccharide molecules. In this case, the release of the drug is due to the action of enzymes that can break the bond between two monomers of the linker instead of between the GO and the drug [210]. Trusek et al. [78] attached DOX on the surface of GO through the Gly-Gly-Leu tripeptide linker. The metalloendopeptidase thermolysin enzyme catalyzes the hydrolysis of the peptide bond between Gly and Leu. Consequently, DOX is released with leucine attached to its NH_2_ group. The results also revealed that this molecular modification did not affect the therapeutic properties of DOX, while the release efficiency was satisfied. In another study of the same group [210], the same linker was exploited to attach amoxicillin to GO. A bromelain enzyme was added to the solution at concentrations of 0.04, 0.10, 0.20, and 0.40 mg/mL and the release efficiency measured was around 19% for the lower dose, 62% for the next dose, and reaching 90% within 24 h for the two higher doses.

Although drug delivery is highly dependent on the enzyme concentration achieved by hydrolysis of the specific links, high efficiency values can be achieved. Regarding the use of this approach for the controlled release of genes, the formation of permanent linkages between linkers and nucleotides could compromise the function of the gene itself. It could be further exploited as an intermediate link between GO and another molecule capable of interacting with the gene, or by encapsulating the nucleotide within biomaterials that undergo enzymatic degradation. The possibility of encapsulating RNA and DNA within the gelatin hydrogel has been attested [211,212]. Once internalized into cells, gelatin has been shown to undergo degradation in the endo-lysosomal complex, which contains many types of hydrolases (i.e., phosphatases, cathepsins), including collagenase, to which gelatin is highly sensitive [194]. The promising results obtained in studies on controlled drug delivery through the intervention of enzymes could be an encouragement to assess this possibility for the transport of genes.

### 4.4. Near IR Stimuli and Treatment

In recent years, GO has also attracted a lot of attention as a photothermal sensitizer nanomaterial. GO in fact holds great promise for its ability to convert absorbed light into localized heat through plasmon resonance, taking advantage of its wide absorption spectrum ranging from UV to NIR [195,213]. GO was explored in cancer phototherapies, mainly including photothermal therapy (PTT) and photodynamic therapy (PDT), upon specific light irradiation on the cancer site. Table 7 reports some recent GO-based studies using NIR against breast cancer. These approaches are particularly advantageous because the specific targeting of cancer cells with high selectivity reduced the side-effects compared to the undesired treatment of healthy cells [168]. Zhang et al. [169] suggested that a significant increase in temperature was found to be proportional to the GO concentration and the applied irradiation. The maximum temperature was reached at approximately 48−50 °C. Such temperatures proved to be sufficient to generate a heat shock in the targeted cancer cells capable of blocking their rapid growth.

Moreover, it was also shown that the photothermal conversion of GO could increase its payload release efficiency, allowing a multimodal treatment of cancer cells using NIR radiation. Gadeval et al. [195] studied GO nanoparticles reduced and stabilized by quercetin and functionalized with FA as a therapy against triple-negative breast cancer (i.e., MDA-MB-231 cells). In addition to a PTT effect, the nanocarrier showed a pH-dependent and NIR-dependent quercitin release trend. At acidic pH (5.5), higher release values (35% in 48 h) were obtained than at physiological pH (7.4; 20% in 48 h), while after NIR irradiation, the release rates were significantly higher for both pH values (40% at pH 7.4 vs. 70% at pH 5.5). This phenomenon can be explained by the increased kinetic energy generated by the NIR laser irradiation. By increasing the kinetic energy, the vibrations of the atoms increase, which in turn induces an easier breakage of the π-π stacking between GO and quercitin, resulting in the release of the payload [195]. Comparable results were also obtained by Roy et al. [136], whereby rGO was functionalized by the modified poly(allylamine hydrochloride (MPAH) and FA. This nanocarrier showed that the release of pDNA from the nanocomposites in the solution increased with NIR irradiation (35%) compared to without NIR irradiation (5%), as observed by the increased fluorescence intensities of the supernatants collected. The rGO can represent an interesting solution for the cellular transfection process due to its higher affinity for the cell membrane and higher cellular uptake. In addition, rGO has a higher conductivity that enhances its photothermal response upon light absorption in the NIR range. This characteristic could, therefore, be an advantage in terms of the endosome escape, controlled release of the therapeutic agent, and PTT and PDT treatments of cancer. Nevertheless, rGO is characterized by a lower density of functional groups on its surface, which can be detrimental from the point of view of the drug/gene loading efficiency, affecting the transfection results. Moreover, a higher toxicity of rGO was recorded compared to GO under the similar experimental conditions, which may limit its application in further transfection steps.

As previously mentioned in Section 2.2, Assali et al. [62] studied a multimodal GO-based nanoparticle functionalized with gold nanorods, PEG as a stabilizer, and PLA as a targeting agent to treat breast cancer cells (i.e., MCF-7, MDA-MB-231 and HU-02) with miRNA-101 and NIR thermal therapy. NIR radiation was used to induce a photothermal effect combined with an enhanced gene release in cancerous breast cells. The results showed low breast cancer cell viability (<20% in 72 h, Figure 9) when performing NIR radiation with or without miRNA. In addition, the NIR laser showed no toxicity on normal cells. 

Zeng et al. [88] exploited GO’s responsiveness to NIR radiation to develop a nanocarrier system for the joint delivery of genes and drugs to solve the big challenge of the drug resistance for breast anticancer treatment. The system was composed of an FA-conjugated, PEI-modified PEGylated graphene (PPG-FA/siRNA/Dox) for the dual delivery of DOX and siRNA. The heat generated by the PPG-FA was analyzed to confirm the actual ability of the system to convert light energy into thermal energy. The results confirmed that the temperature increased over time to a maximum of about 43 °C after 10 min for NGO concentrations of 15 μg/mL. Therefore, despite the numerous functionalization strategies, the high NIR absorbance of the GO allowed the support of a remarkable photothermal ability. Furthermore, the release of DOX was studied at two different pH levels (7.4 and 5.0), with and without NIR radiation. 

As previously demonstrated by Gadeval et al. [195], the lowest release rate was recorded at physiological pH and without radiation (about 20% in 48 h). In the presence of acidic pH or NIR, the release values increased, reaching a maximum with the combination of these two stimuli (about 60% in 48 h). Finally, the presence of siRNA confirmed the ability to efficiently transport the gene, as the drug resistance effect was significantly decreased compared to the DOX-only control.

### 4.5. Heat Treatment

In addition to the high acidity and imbalance in the levels of proteins and intracellular enzymes, the tumor environment was reported to present a higher temperature of about 2−5 °C more than the physiological temperature [206]. This characteristic represents another stimulus that can be exploited to engineer nanoparticles for the controlled release of drugs and genes in the pathological site. 

GO has exhibited high thermal conductivity (i.e., about 5300 W/mK at room temperature) [184]. Other materials known for their sensitivity to temperature changes are thermo-responsive polymers, such as poly(N-isopropylacrylamide) (PNIPAM), with a phase transition temperature of around 32 °C [197]. It can be combined with natural polymers, such as chitosan or HA, to form smart hydrogels with a similar transition temperature [198]. When these polymers were linked to GO, the volume phase transition temperature (VPTT) of the polymers decreased due to the high heat conductivity of GO. For instance, the PVA/PNIPAM hydrogel normally exhibits a VPTT of around 35 °C, but when combined with GO, this temperature decreases to 34 °C [184]. In this case, the hydrogel exhibited an excellent temperature response due to the increased sensitivity caused by the rapid thermal conduction of the GO. Another plausible reason is that the presence of GO induces the formation of a denser hydrogel network, leading to low adsorption of water. As a result, the material would have a lower swelling ratio and a higher volume change in response to temperature variations [184]. Most of the published efforts were focused on building smart hydrogels to realize a nanoscale system that can tune their properties once they reach the tumor to release the payload.

Wang et al. [183] prepared a thermo-sensitive delivery system by functionalizing the 2D GO flakes with PNIPAM-polyehylene oxide (PEO) through 1-pyrenebutyric acid N-hydroxysuccinimide ester. The GO complex was previously loaded with Adriamycin, an aromatic drug used for treating tumors. Polymer chains form hydrogen bonds with water when the temperature is above the specific critical solution temperature (LCST). Once the temperature is lower than the LCST, the hydrogen bonds break and the material shrinks, releasing the drug from the GO-based carriers [197]. 

Although there are few studies using this approach for gene transport, we found that GO-based hydrogels were exploited for drug/gene delivery. With an LCST lower than body temperature, such temperature-sensitive smart hydrogels were used to make a gel containing GO flakes. During the preparation at room temperature, they were kept in liquid form and could be injected in a mini-invasive way in the tumor site. At 37 °C, the hydrogel tended to go through gelation, supporting the accumulation of GO carriers in the target site and their sustained release over time in accordance with the degradation time of the hydrogel [216]. Fong et al. [198] fabricated a GO nanoparticle decorated with FA to load and deliver DOX to MCF-7 breast cancer cells. Subsequently, these nanocarriers were loaded into a thermo-sensitive hydrogel made of hyaluronic acid–chitosan-g–PNIPAM that presented an LCST range of around 30−32 °C. This material allowed the injection and accumulation of nanoparticles into the tumor site due to the in situ formation of the hydrogel in response to the higher temperature, supporting their release concomitant with its degradation. A similar approach was explored for the gene delivery [216].

### 4.6. Competitive Molecules

Another approach for GO-based gene delivery in targeting breast cancer cells is through a mechanism of competition between the oligonucleotide and other molecules. Inside cells, the given specific molecules that have high affinity towards GO tend to bind together to reach a more stable form. As a result, the payload associated with GO could be “replaced” with these molecules on the graphene surface. This approach would form GO nanocarriers without additional functionalization, leading to a limited fabrication step, which may increase the reproducibility and make them non-toxic [117]. The use of molecules that compete for the binding to the GO surface may be a solution to induce the release of genes from graphene flakes. These molecules include small polyaromatic molecules, such as methylene blue (MeB) and rhodamine 6G (R6G). They possess extended aromatic systems that allow the formation of π-π stacking interactions with graphene, similar to that of DNA [187,188]. Therefore, they have the potential to competitively desorb DNA molecules bound on the GO surface. Another possibility could be to exploit the cellular DNA. A single-stranded DNA molecule loaded on the GO could be influenced by its higher affinity with the DNA in the cell, which would then compete with the GO for binding of the strand complementary to itself [187]. These approaches are schematically illustrated in Figure 10.

These mechanisms have been tested by Chen and Zhang [187], who loaded oligonucleotides and ss-DNA onto SWNTs. Binding occurred through a mechanism of wrapping DNA around the nanotubes, as DNA possesses a negatively charged chain due to sugar-phosphate bonds, which can impart a large degree of hydrophilicity to the SWNTs by preventing them from aggregating in aqueous solutions. Furthermore, such binding was facilitated by the tendency of the four nitrogenous bases of DNA to form attractive π-π stacking interactions due to their extended π structures. MeB and R6G were used as competitive molecules and compared with creatinine and fluorescein to understand the key feature of achieving efficient competition with GO. Interestingly, amino and amino groups on creatinine are similar to those found in R6G and MeB, which are positively charged. However, creatinine does not have an extended aromatic structure, whereas fluorescein has the opposite characteristics. The results showed that the extended aromatic structure was a key feature, with the positive charges facilitating the process. In this study, the results obtained through gel electrophoresis confirmed the formation of the double-stranded DNA (dsDNA) from the ssDNA loaded on the GO. In this case, the driving force for the release of the oligonucleotide from the nanocarrier was probably the affinity between the strands. The possibility of creating hydrogen bonds with the complementary nitrogenous bases was also thermodynamically favored with respect to the π-π stacking interactions with graphene.

The same approach was also evaluated by Hsieh et al. [188], in which the dye R6G was used to visualize and induced DNA release from GO nanocarriers. In addition to competing for bonds on GO, rhodamine is pH-sensitive and protonated under acidic conditions, emitting fluorescence at around 588 nm. Therefore, R6G was used to stimulate the controlled release of the dT30 gene when GO nanocarriers were internalized into RAW 264.7 cells. Simultaneously, the binding formed between rhodamine and the graphene surface led to the quenching of R6G’s fluorescence by GO, which was used to visually confirm the actual internalization of graphene nanoparticles and their gene release within the cell.

In addition, directly linking genes to the GO surface was explored using non-covalent bonds. Although some difficulties related to this mechanism can be found in the literature [160], several studies have demonstrated the possibility of achieving such direct binding [160,187,188]. On the other hand, it was pointed out that once the gene was linked on GO, it showed almost no tendency to separate from the graphene surface, demonstrating stable binding but discouraging its application for gene therapy [160]. Therefore, in the case of loading genes on GO through π-π stacking interactions, applying an external driving force for the release can be beneficial.

The possibility of using an external molecule could represent a potential approach. Nevertheless, there are no recent studies in this field, probably due to the difficulties encountered in the procedure of loading and releasing genes in the absence of further functionalization [160]. Either way, the results obtained in these studies seem promising, although it would be necessary to extend these studies to highlight the release efficiency on tumor cell lines to understand the true potential of this mechanism.

## 5. Challenges and Perspectives

Graphene-oxide-based nanocarriers have shown great potential in gene transfection and could be used in breast cancer treatments by overcoming the limitations and side effects of chemotherapy. This carbon-based material allows a facile surface modification to enhance its biological application and gene delivery efficiency. Although GO is controllable in terms of its lateral size, achieving a narrow size distribution with cost-effective and less time-consuming approaches remains a challenge. As-synthesized GO flakes without size control (ranging from microns to nanoscale) highlight the importance of reaching a uniform size of GO flakes for drug/gene delivery purposes to facilitate passing through the barriers.

Polymer–cationic agent modifications have been widely exploited to create GO carrier formulations. Nevertheless, the possibility of enhancing the biocompatibility and stability in the absence of cationic agents is yet to be investigated. In most studies, nucleic acids were primarily loaded on a polymer, dendrimer, or peptide-modified GO, while fewer studies have focused on non-covalent interactions between genes and GO. In addition, studies have widely shown the successful delivery and release of siRNAs and miRNAs with the purpose of gene silencing through binding these genes to mRNA, while fewer studies have reported on the delivery of plasmids that can internalize nuclei of the cell, causing specific gene expression in breast cancer cells. 

Since versatile factors affect the journey of GO-based carriers in the body, it is crucial to reach a comprehensive understanding of GO’s characteristics. Despite the contradictions regarding the toxicity of GO in the literature, it should be noted that the function and toxic effect of GO in biological applications are profoundly affected by the synthesis route, purification, post-processing, size distribution, shape, functionalization, charge, dose, and time of exposure to biological compartments. The production of GO is one key aspect impacting its performance. Although modifying the synthesis conditions can increase the production yield, post-processing approaches play a complementary role in achieving a stable, consistent, and reproducible material. The methods, including exfoliation, purification, and size separation, can be further improved to be less time-consuming than conventional methods (e.g., ultrasonication, centrifugation, and filtration), as reported in the literature. In other words, by modifying the post-synthesis methods, not only can a uniform material be obtained, but also a larger quantity of GO can be processed efficiently, which at the same time will have consistent features. It is also noteworthy that although successful in vitro and in vivo results regarding gene transfection using GO-based carriers have been reported, the lack of consensus in the research findings raises the need to better understand the characteristics of this material prior to clinical investigation.

In addition, the capability of GO to respond to various stimuli, including pH, the reducible intracellular environment, enzymes, NIR irradiation, and the presence of competitive molecules, allows the engineering of smart drug/gene carriers with controlled release for cancer therapy. Several studies can be found in the literature regarding the exploitation of pH and NIR radiation for the controlled release of genes from GO carriers for breast cancer treatment. Other stimuli such as redox, enzyme, and heat treatments, however, have not been fully assessed for this application so far, although several examples were exploited for drug release.

Moreover, some of these stimuli, such as pH and temperature, do not directly act on the GO but rather on moieties that can be efficiently loaded on it. This may have led to the contradictory findings in the literature, as the efficiency of cargo release would unlikely depend on the GO but rather on the functional molecules used. Therefore, it is clear that additional challenges have to be overcome to achieve an adequate gene delivery system, such as accomplishing the efficient loading of oligonucleotides onto the GO and their controlled release into the right cellular compartment. This applies especially to the transport of DNA segments. In the case of efficient transfection, GO must be able to penetrate inside the cell nucleus to release the gene onsite. For this purpose, these approaches could represent potential mechanisms to increase the transfection efficiency in target cancer cells.

## Figures and Tables

**Figure 1 ijms-23-06802-f001:**
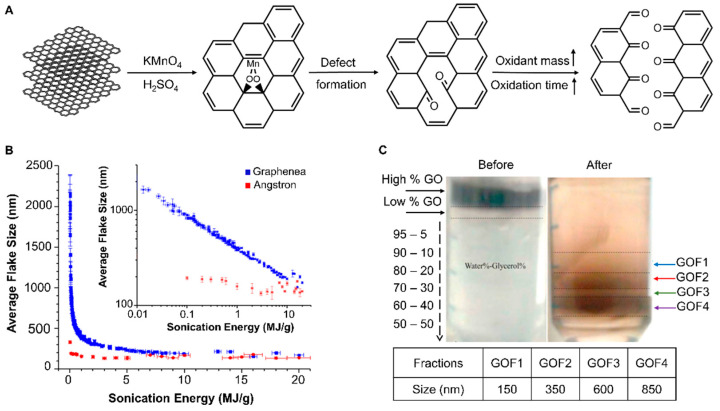
Approaches to reduce the lateral size of GO through (**A**) increasing the oxidant mass and the time of oxidation during synthesis, which causes ruptures in defected regions of GO flakes. The figure is adapted from [33,58] with open access permission. Copyright © 2020 Tufano, Vecchione, and Netti. (**B**) Increasing the ultrasonication time and energy, leading to a decrease in the size of GO from 2 µm to 150 nm. The figure is reprinted with permission from [58]. Copyright 2017 American Chemical Society. (**C**) Centrifugation and fractionation of 0.01 mg/mL GO in water–glycerol mixture at 3000× *g* for 1 h. The figure is reprinted with permission from [59] Elsevier.

**Figure 2 ijms-23-06802-f002:**
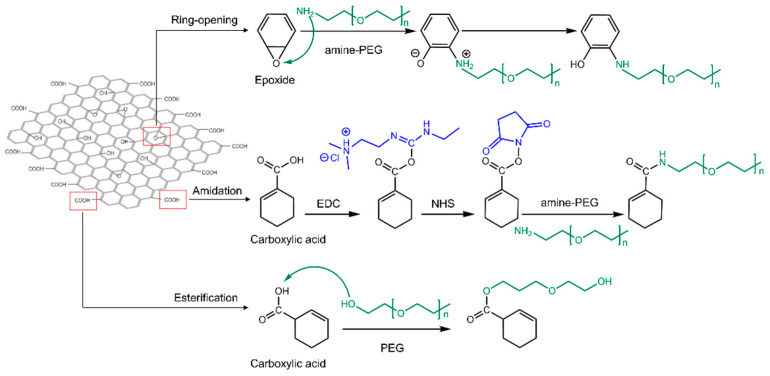
Commonly used modification methods to functionalize GO for drug/gene delivery applications, including the ring-opening of epoxide, amidation using EDC-NHS chemistry, and esterification in the absence of NH_2_ in carboxylic acid regions.

**Figure 3 ijms-23-06802-f003:**
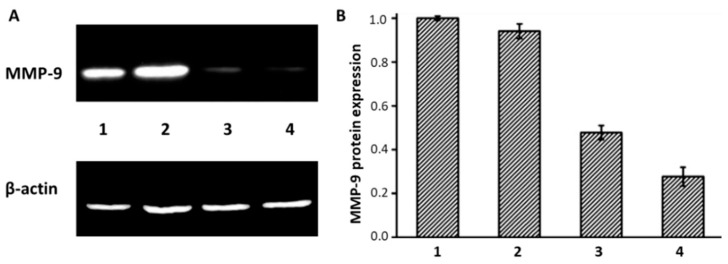
MMP-9 protein expression determined by Western blot analysis (**A**) and intensity analysis of (**B**) MMP-9 expression as the ratio of MMP-9 to β-actin from Western blot results: (1) PBS; (2) GO-PAMAM; (3) GO-PAMAM/MMP-9 (weight ratio of 10:1); (4) PEI-25k/MMP-9 (weight ratio of 4:3) [87]. Figures are reprinted with permission from Elsevier.

**Figure 4 ijms-23-06802-f004:**
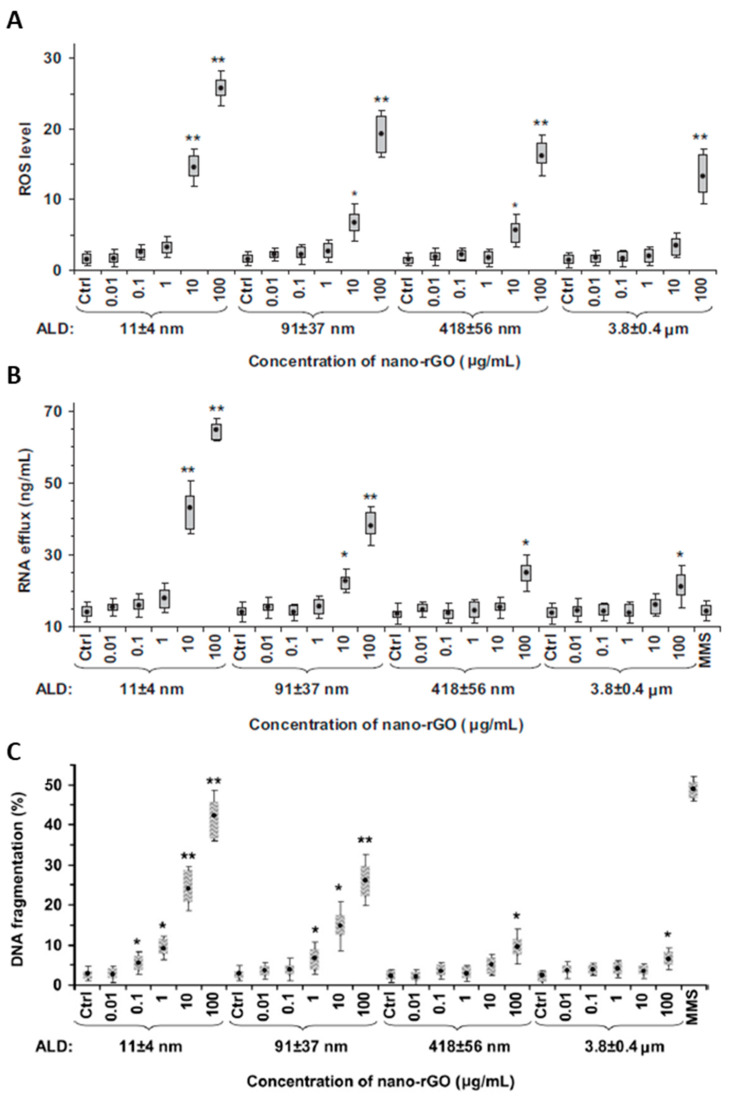
(**A**) ROS-generation-, (**B**) RNA-efflux-, and (**C**) DNA-fragmentation-induced toxicity of rGO flakes of different sizes (ALD: average lateral dimension; *: *p* < 0.05, **: *p* < 0.01) [112]. Figures are reprinted with permission from Elsevier.

**Figure 5 ijms-23-06802-f005:**
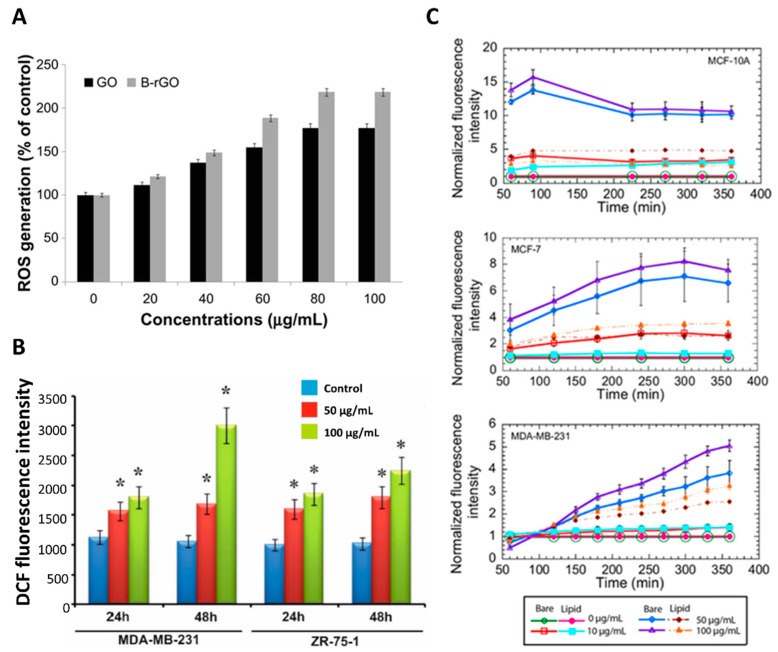
The effect of GO on the production of ROS in breast cancer cells. (**A**) Generation of ROS in MCF-7 cells induced by GO and bacteria-reduced GO (B-rGO) (treated groups showed statistically significant higher ROS production with respect to the control (*: *p* < 0.05)). The figure is adapted from [147], copyright 2013 Gurunathan et al., originally published by and used with permission from Dove Medical Press Ltd. (**B**) ROS production induced by rGO in MDA-MB-231 and ZR-75-1 cells with different concentrations of nanoparticles. The figure is adapted with open-access permission from [118], copyright 2021 Kretowski et al., publisher and licensee MDPI. (**C**) Oxidative stress caused by different concentrations of rGO and measured through DCFDA assay on MCF-7 and MDA-MB-231 breast cancer cells and MCF-10A normal breast cells. Figures are reprinted and adapted with permission from [119], copyright 2020 American Chemical Society.

**Figure 6 ijms-23-06802-f006:**
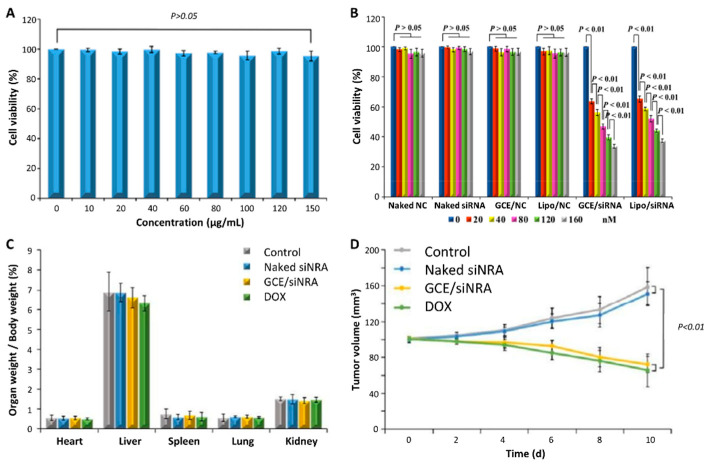
Cytotoxicity of GO and antiproliferation effect of GO-siRNA in vitro and in vivo. (**A**) Cytotoxicity of GCE and (**B**) antiproliferation of GCE-siRNA on MCF-7 cells in vitro. (**C**) Cytotoxicity on normal tissues and (**D**) on breast cancer tumor of GCE-siRNA in vivo. Figures are reprinted with open access permission from [64], copyright 2021 Shenyang Pharmaceutical University, published by Elsevier.

**Figure 7 ijms-23-06802-f007:**
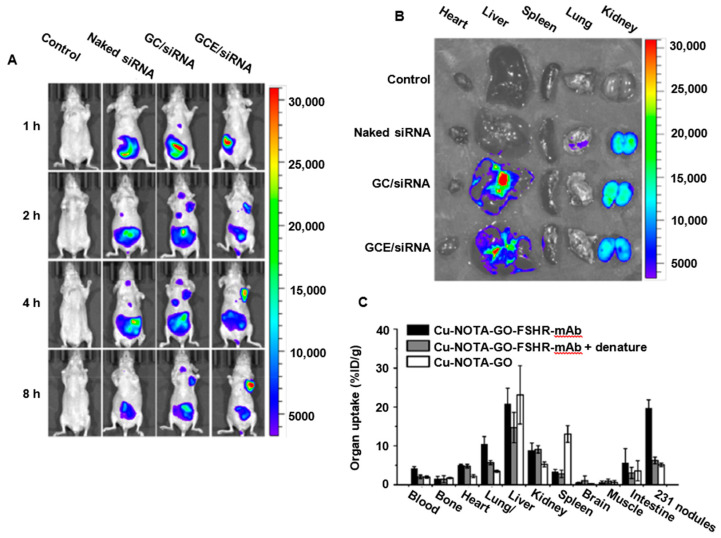
In vivo human breast cancer xenograft model in nude mice to observe the Survivin-siRNA biodistribution via fluorescence imaging. (**A**) Images taken at 1, 2, 4, and 8 h after injection of all formulations. (**B**) Fluorescent images obtained after dissection of the main organs from each formulation group. Figures in (**A**,**B**) are reprinted with open access permission from [64], copyright 2021 Shenyang Pharmaceutical University, published by Elsevier. (**C**) Biodistribution of Cu-labeled GO nanocarriers evaluated in ex vivo mice organs after 24 h of intravenous injection (NOTA: 2-S-(4-isothiocyanatobenzyl)-1,4,7-triazacyclononane-1,4,7-triacetic acid). Figures are reprinted with permission from [178], copyright 2016 Elsevier.

**Figure 8 ijms-23-06802-f008:**
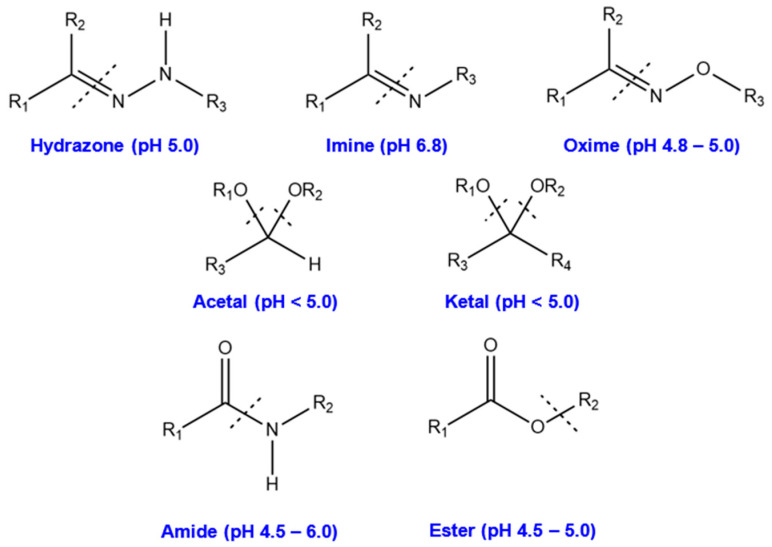
Acid-sensitive cleavable linkers used in designing smart nanoparticles [97,192]. Figures are reprinted with permission from Elsevier.

**Figure 9 ijms-23-06802-f009:**
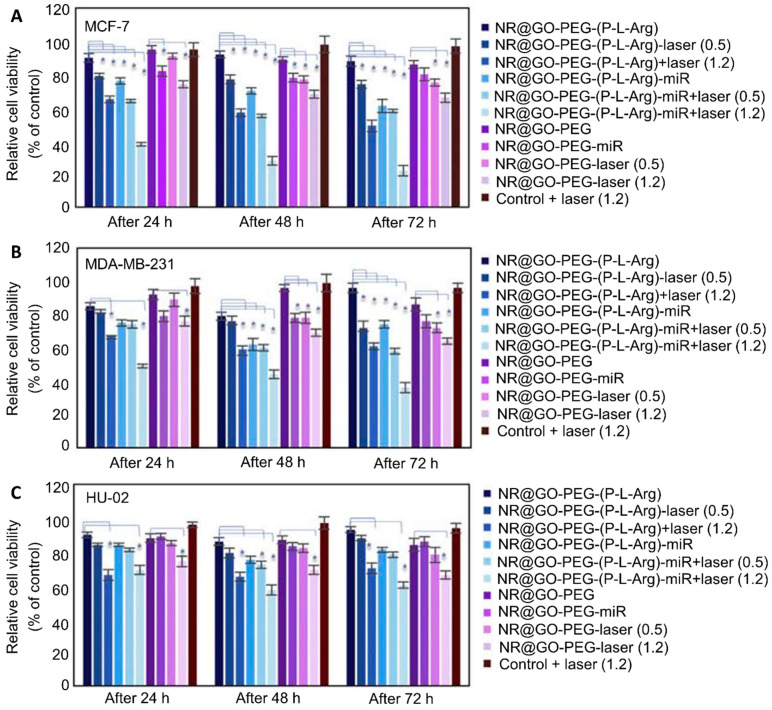
Cell viability of (**A**) HU-02 primary cell line and (**B**) MDA-MB-231 and (**C**) MCF-7 breast cancer cell lines. The results show the vitality of the cells treated with GO and GO-miRNA nanocarriers with respect to the control and exposed or not to laser irradiation (0.5 and 1.2 W/cm^2^) for 10 min, *: *p* < 0.05. The figures are reprinted with permission from Elsevier [62].

**Figure 10 ijms-23-06802-f010:**
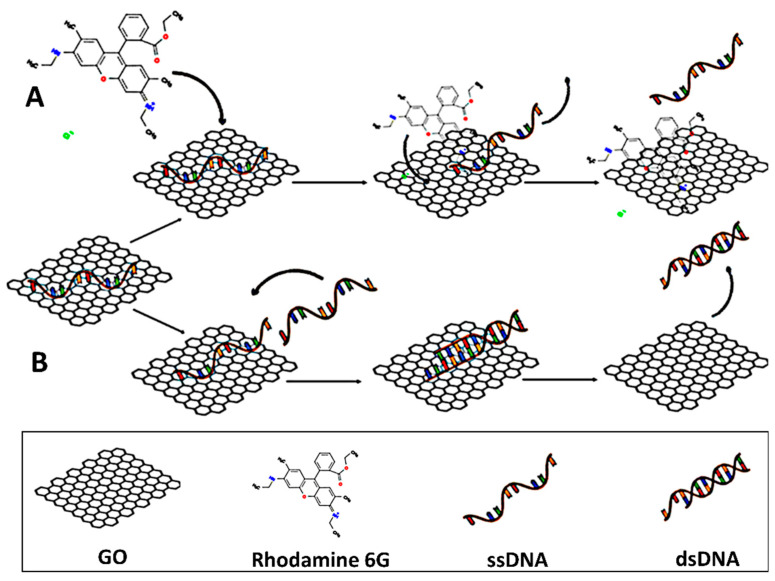
Schematic representation of the gene release from GO through the association of (**A**) competitive molecules (rhodamine 6G is reported as an example) and (**B**) ssDNA.

**Table 1 ijms-23-06802-t001:** Graphene oxide modifications with the objective of gene transfection.

GO Modification	Modification Mechanism	Agents
Covalent	Amidation	PEG [62,63,77,86]
		PAMAM [16,17,87]
		PEI [88,89]
		Chitosan [64,65,66]
		DEX [90]
	Ring-opening reaction	PEG [17]
		APTES [53]
	Esterification	PEG [91]
Non-covalent	π-π stacking	MUC1 aptamer [72]
	Electrostatic interactions	PEG [67,70]
		PEI [67]
	Hydrophobic interactions	Phospholipids [52,70]

**Table 2 ijms-23-06802-t002:** The effects of the surface functionalization and N/P ratio of GO-based carriers on gene silencing in breast cancer cells.

GO Modification	Payload Type (Targeting Agent)	N/P Ratio (*w*/*w*)	Gene Silencing	Cell Type
PEG	^a^ miRNA 101 (PLA) [62]	1−3.8	-	MCF-7, MDA-MB-231
	^a^ cd-siRNA (R8) [70]	10	-	
	^a^ P-gp siRNA (FA) [88]	4	70%	
	^b^ Rictor siRNA (R8) [67]	0.5	70%	
	^a^ EPAC1-siRNA [17]	2	62%	MDA-MB-231
PAMAM	^a^ EPAC1-siRNA [17]	2	62%	MDA-MB-231
	^b^ miR-21i [16]	-	-	
	^b^ MMP-9 shRNA [87]	10	52%	MCF-7
PEI	^a^ P-gp siRNA (FA) [88]	4	70%	MCF-7, MDA-MB-231
	^b^ Rictor siRNA (R8) [67]	0.5	70%	
Chitosan	^b^ Survivin siRNA (anti-EpCAM) [51]	30	44%	MCF-7
	^a^ HIF-1α siRNA (HA) [65]	6.5	62%	4T1
	^a^ EGFR siRNA (FA) [66]	-	-	MCF-7, MDA-MB-231
Peptide	^a^ cd-siRNA (R8) [70]	10	-	MCF-7, MDA-MB-231
	^b^ Rictor siRNA (R8) [67]	0.5	70%	
	^b^ Survivin siRNA (anti-HER2) [64]	40	50%	MCF-7

^a^ in vitro; ^b^ in vitro and in vivo.

**Table 3 ijms-23-06802-t003:** Dose- and time-dependent cytotoxicity of GO-based nanocarriers in different cell lines.

Nanoparticle	Cell Type	Dose (µg/mL)/Dependency		Time (Day)/Dependency	
GONR (PEG-DSPE) [122]	HeLa; MCF-7; SKBR; NIH3T3	10−400	Y	0.5−2	Y
GO	PMO; J774A.1; LLC; MCF-7; HepG2; HUVEC [117]	0−20	Y	1, 2, and 4	-
	HDF [129]	5, 10, 20, 50, and 100	Y	1, 2, 3, 4, and 5	Y
	A549 [130,131]	10, 25, 50, 100, and 200	N	1, 2, and 3	N
	HepG2 [132]	4, 8, and 16	Y	1 and 3	Y
GO, GOD and GOP ^a^ [114]	MCF-7	2.4, 24, and 48 (μg/cm^2^)	Y	1, 2, and 3	N
GO-DEX-Apt-CUR ^b^ [90]	4T1; MCF-7	Up to 300	Y	1 and 2	N
Lipid-rGO [119]	MDA-MB-231	10, 50, and 100	Y	1 and 2	Y
	MCF-7; MCF 10-A		N		N
rGO [118]	MDA-MB-231; ZR-75-1	25−300	Y	1 and 2	Y
	MCF-7; Hs 578T; T-47D		N		N
Graphene/SWCNT [128]	PC12	0.01, 0.1, 1, 10, and 100	Y	1−24 h	Y

^a^ GO modified with DAB-AM-16 and PAMAM dendrimers. ^b^ Dextran, ssDNA aptamer, and curcumin.

**Table 4 ijms-23-06802-t004:** Size and zeta potential values of GO-based nanocarriers used in cytotoxicity studies.

Nanocarriee	Size (nm)	Zeta Potential (mV)	
		GO	NP
GO-PEI [133]	250−500	−53	+50
GO-DOTAP [52]	100−150	−30	+15
rGO-MPAH-FA [136]	100−230	−45	+40
GO-anti HER2-R8 [51]	120−260	−49	+26
GO-anti EpCAM-Chitosan [64]	70−350	−42	+38
GO-PEG-PAMAM [17]	54−220	−30	+10

**Table 5 ijms-23-06802-t005:** Release mechanisms of GO nanocarriers for gene delivery.

Release Trigger	Responsive Molecules	Responsive Functional Groups/Bonds	Mechanism
pH	GO + cationic molecules, pH-sensitive hydrogels	Amide [162], imine [189], hydrazone [190], ester [191], and oxime [192]	Protonation
Reducible intracellular environment	GO + molecules containing disulfide bonds	Disulfide bonds	Redox [123,185,193]
Enzymes	Phosphatase, collagenase, cathepsin	Peptide bonds	Degradation [186,194]
NIR irradiation	GO	Aromatic C-C bonds	PDT and PTT [168,195,196]
Temperature treatment	Thermo-sensitive hydrogels	Hydrogen bonds between the polymer molecules and/or water	Phase-volume transition [197,198]
Competitive molecules	GO + polyaromatic cationic molecules	Hydrophobic and electrostatic interactions between GO and the polyaromatic cationic molecules	Desorption of genes [187,188]
	GO + cellular DNA	Hydrogen bonds between nitrogenous bases of the two DNA strands	Desorption of genes [187]

**Table 6 ijms-23-06802-t006:** The pH-sensitive platforms using GO nanocarriers for drug and gene delivery in breast cancer cells.

pH	Efficiency	GO Functionalization	Drug/Gene	Time (h)	Cell Type
<7	20%	GO-PAMAM [87]	DOX	144	MCF-7
7.4	17%				
<7	40%	PPG-FA ^a^ [88]	DOX	48	MCF-7/ADR
7.4	10%				
<7	60%	CS-g-PMAA/GO ^b^ [202]	DOX	220	MCF-7
7.4	25%				
<7	71%	GO [200]	DXR	30	-
7.4	11%				
<7	95%	FA-rGO/ZnS:Mn ^c^ [203]	DOX	70	MDA-MB 231
7.4	150%				
<7	28%	GO-Gel-BSA [186]	DOX	-	MCF-7
7.4	10%				
<7	58%	CGO-TMC-HA ^d^ [65]	Dinaciclib	36	4T1
7.4	50%				
<7	90%	Chitosan [64]	Survivin_siRNA	200	MCF-7
7.4	30%				
<7	80%	Chitosan-HA [65]	HIF-1α_siRNA and Dinaciclib	32	4T1, CT26, B16-F10, TM3
7.4	70%				
<7	12%	PAMAM [17]	siRNA	72	MDA-MB-231
7.4	61%				

^a^ FA-conjugated high molecular weight branched polyethyleneimine modified PEGylated nanographene; ^b^ chitosan-graft-poly(methacrylic acid); ^c^ chemically reduced graphene oxide combined with manganese-doped zinc sulfide quantum dots and functionalized with folic acid; ^d^ carboxylate graphene oxide (CGO) conjugated with trimethyl chitosan (TMC) and hyaluronate (HA) nanoparticles.

**Table 7 ijms-23-06802-t007:** Optical treatments for breast cancer using GO materials.

Trigger	NPs	Cell Type	Effect
430 nm laser (100 J/cm^2^)	GO-Ag	MCF-7	ROS production [214]
NIR	GO-FA	MDA-MB-231	PTT [195]
	nGO-CuSQDs	MCF-7	Apoptosis [148]
	rGO-nZVI	MCF-7	PTT [215]
LED (660 nm)	GO-MB	MDA-MB-231	PDT [168]
	GQDs-MB	MCF-7	ROS production and apoptosis [213]

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
