# Peer review of "Characteristics of Graphene Oxide for Gene Transfection and Controlled Release in Breast Cancer Cells"

_ijms, 2022, doi:10.3390/ijms23126802_

Round 1

Reviewer 1 Report

In this review the authors investigate on Graphene-based nanomaterials for gene transfection and gene transfection breast cancer treatment. Recently another group has also been published very similar review.

https://doi.org/10.1186/s12951-021-00902-8

However, the review is very informative. Therefore, I would recommend publishing this article after addressing few minor corrections. 

Discuss the comparative biocompatibility of Graphene oxide and reduced Graphene oxide for breast cancer treatment as well as gene transfection.

Author Response

We thank the reviewer for the comments and suggestions. The following changes are made: 

(1) We added the referred review paper as [174];

(2) We added the discussion on biocompatibility of rGO and GO, as well as their comparison for cancer treatment and transfection on pages 14, 18-19, and 31. Added details are highlighted in yellow and listed below 

Page 14:

These results proved that rGO is more cytotoxic than GO, demonstrating a strong interaction formed between graphene and cells in a limited period of time. Since toxicity depends on the physicochemical properties of the materials, the use of reducing agents to deoxygenate GO can lead to the formation of materials with different characteristics (i.e., particles size, density of functional groups on the surface), and thus different toxicity. In fact, following some reduction treatments, rGO has been shown to aggregate due to van der Waals interactions and exhibited different sizes. This characteristic is known to be a major cause of increased toxicity, as will be discussed in the following sections.

Page 18-19

This could indicate that non-functionalized rGO is not an adequate material for breast cancer treatment, as such high ROS concentration could result in an increased risk of mutations of normal breast cells into tumor tissue. Even when tested in lower doses than GO, rGO was shown to induce the same or higher levels of cell death, depending on the time and dose of exposure to cells. This phenomenon can be related to the higher affinity of rGO to the cell membrane that increases the hydrophobic interactions between them. In addition, some chemical agents used for GO reduction can generate metallic impurities and organic contamination, causing alterations in the interaction with cells, leading to cell membrane damage and apoptosis.

Page 31:

Comparable results were also obtained by S. Roy et al. [136], in which rGO was functionalized by the modified poly(allylamine hydrochloride (MPAH) and FA. This nanocarrier showed a release of pDNA from the nanocomposites in the solution increased with NIR (35%) compared to without NIR (5%), as observed by the increased fluorescence intensities of the supernatants collected. rGO can represent an interesting solution for cellular transfection process due to its higher affinity to the cell membrane and a higher cellular uptake. In addition, rGO has a higher conductivity that enhances its photothermal response upon light absorption in the NIR range. This characteristic could therefore be an advantage in terms of endosome escape, controlled release of the therapeutic agent, and PTT and PDT treatment of cancer. Nevertheless, rGO is characterized by a lower density of functional groups on its surface that can be detrimental from the point of view of the drug/gene loading efficiency, affecting the transfection results. Moreover, higher toxicity of rGO was recorded compared to GO under the similar experimental conditions, which may limit its application in further transfection steps.

Reviewer 2 Report

In this review, the authors described the characteristics of GO and functionalized GO as platforms for nucleic acid delivery and their properties for pharmaceutical and biomedical applications.

The manuscript is well-written and organized. Extensive literature review is included. The figures are impressive. I believe that this manuscript is very useful for the scietists in the field of nanomaterials.

My comments are:

1. The methods of the literature review is missing. The search keywords should be presented. 

2. The future perspectives about the scale up of GO materials.

Author Response

We very much appreciate the reviewer's comments and suggestions. The following changes are made: 

We added the discussion on scaling up and the edit is highlighted in yellow and listed below. 

Page 35-36:

Since versatile factors affect the journey of GO-based carriers in the body, it is crucial to reach a comprehensive understanding of GO characteristics. Despite contradictions regarding GO toxicity in the literature, it should be noted that the function and toxic effect of GO in biological applications are profoundly affected by synthesis route, purification, post-processing, size distribution, shape, functionalization, charge, dose, and the time of exposure to biological compartments. The production of GO is one key aspect impacted its performance. Although modifying synthesis conditions can increase production yield, post-processing approaches play a complementary role in achieving a stable, consistent, and reproducible material. The methods, including exfoliation, purification, and size separation, can be further improved to be less time-consuming than conventional methods (e.g., ultrasonication, centrifugation, and filtration), as reported in the literature. In other words, by modifying post-synthesis methods, not only a uniform material can be obtained, but also a larger quantity of GO can be processed efficiently, which at the same time has consistent features. It is also noteworthy that although successful in-vitro and in-vivo results regarding gene transfection using GO-based carriers have been reported, the lack of consensus in research findings raises the demand to better understand the characteristics of this material prior to clinical investigation.